

# Melt-enhanced strain localization and phase mixing in a large-scale mantle shear zone (Ronda peridotite, Spain)

Sören Tholen[1], Jolien Linckens[1,2], Gernold Zulauf[1]

[1] Institut für Geowissenschaften, Goethe Universität Frankfurt a.M., Altenhöferallee 1, D-60438, Germany
[2] Tata Steel, R&D, 1970 CA Ijmuiden, The Netherlands

*Correspondence to*: Sören Tholen (tholen@geo.uni-frankfurt.de)

**Abstract.** Strain localization in upper mantle shear zones by grain size reduction and the activation of grain size sensitive deformation mechanisms (grain boundary sliding, diffusion creep) is closely linked to phase mixing. With its mylonitic grain sizes (50-100 µm) and well mixed phase assemblage, the km-scale shear zone at the northwestern boundary of the Ronda

peridotite is in this respect no exception. In transects across the "mylonitic" into the less deformed "tectonic" part of this shear zone four dominant microstructural domains were identified: olivine-rich matrix, mixed matrix and neoblast tails of clino- and orthopyroxene. In these, phase mixing quantities, its formation processes and its impact on strain localization were analyzed by a combined microstructural (EBSD) and geochemical (EPMA) analysis. Dominant microstructure of all samples is the mixed matrix composed of olivine, ortho- and clinopyroxene. Its homogenous distribution of interstitial, and/or wedge-

shaped pyroxenes contradicts mechanical mixing. In general, high (> 60%) phase boundary percentages in all four microstructural domains indicate extensive phase mixing independent from microstructural domain and distance to the deformational center of the shear zone located at the NW boundary of the peridotite massif. The constant grain sizes with local variations independent on the distance the deformational center indicate a broad scale deformation with ± constant stresses in the entire mylonitic area. Decreasing Mg# and increasing Ti contents with increasing distance to the NW shear

zone boundary, highly lobate phase boundaries, homogenous phase mixing and secondary phase distribution in all samples corroborate a metasomatic phase mixing by melt-rock reactions and crystallization of pyroxenes in the entire shear zone transect. Consistent geochemistry and phase assemblage in mylonites and tectonites but a change from equiaxial (tectonites) to wedge-shaped pyroxenes aligned in the foliation (mylonites) indicate a pre- to syn-deformational melt infiltration. Following the geochemical gradient, the potential source of melt is below a structurally deeper "melting" front which

separates the sheared peridotites from coarse granular peridotites. The presence of mixed matrix in the entire shear zone, its microstructural and geochemical consistency indicate that the melt-infiltration was fundamental for the formation of and the strain localization in the major shear zone of the NW Ronda peridotite.



## 1 Introduction

Deformation in the upper mantle is localized in ductile shear zones. Accommodating most of the deformation in the lithospheric mantle, the shear zones have a major imprint on large scale deformation and plate tectonics (Bercovici and Ricard, 2014; Drury et al., 1991). To localize strain, weakening must occur. Weakening in turn is dependent on an initial heterogeneity/anisotropy and a softening mechanism localizing the strain in the area of heterogeneity and later on in the shear zone itself. In the lithospheric mantle, several types of heterogeneity were identified as potential "seeds" for strain localization: Large-scale variations in the geothermal gradient as present for hot plumes or cold lithospheric roots of cratons, major-element and modal heterogeneities as present in the compositional layering of most peridotite massifs, the presence of melt, variations in the hydration state of particularly olivine, grain size heterogeneities and lateral changes of the olivine CPO (e.g., Tommasi and Vauchez, 2015). Strain softening mechanisms that localize and maintain deformation were subdivided into three types (Drury et al., 1991): Thermal softening caused by shear heating and the positive feedback of temperature and strain rate (e.g., Kelemen and Hirth, 2007), geometric softening caused by the anisotropy in creep strength of grains aligned in a crystallographic preferred orientation (CPO) (Passchier and Trouw, 1996) and microstructural or reaction softening which occurs by grain size reduction and the activity of a grain size sensitive deformation process (Drury and Urai, 1989). Both, thermal softening as well as microstructural softening depend on the presence of mixed phase assemblage, either as seed or as stabilization for strain localization (e.g., Kelemen and Hirth, 2007; Linckens et al., 2015). Phase mixing in the upper mantle has been ascribed to several different processes which can either be deformation or reaction induced. Deformation induced phase mixing is commonly associated with grain boundary sliding (GBS). During GBS, neighbour switching of grains was reported to form mixtures (e.g., Boullier and Gueguen, 1975; Hirth and Kohlstedt, 2003) but also aggregates (Hiraga et al., 2013). Furthermore, disaggregation of single-phase domains at high shear strains ("Geometric mixing") was reported by Cross and Skemer (2017). Additionally, nucleation of neoblasts in creep cavitations during GBS leading to phase mixtures was reported by Précigout and Stünitz (2016). Reaction induced phase mixing is bound to either metamorphic (P-T) or metasomatic (melt/fluid) reactions. In the upper mantle, phase transitions from garnet to spinel and to plagioclase peridotites change the phase assemblage and the mineral chemistry of all present phases (e.g., Borghini, 2008). Thereby, metamorphic reactions can lead to phase mixing and, during deformation, to the formation of ultramylonites (Furusho and Kanagawa, 1999; Newman et al., 1999; Tholen et al., 2022). Additionally, the interaction of rock and melt or fluid can cause phase mixing by precipitation of neoblasts and reactions with porphyroclasts/old grains (e.g., Dijkstra et al., 2002; Kaczmarek and Müntener, 2008).

With its decreasing grain size, scattering of pyroxene neoblasts and concurrent diminishing of prior strong olivine CPOs towards the contact to the bordering Jubrique unit Ronda's NW tectonitic/mylonitic zone is commonly interpreted as km-scale upper mantle shear zone (Garrido et al., 2011; Précigout et al., 2013, 2007). In contrast to earlier studies on phase mixing in upper mantle shear zones by the authors where mixing depends on metamorphic and metasomatic reactions (Linckens and Tholen, 2021; Tholen et al., 2022) Ronda's mylonites are thought to have little to no metamorphic or



metasomatic influence (e.g., Johanesen and Platt, 2015; Précigout et al., 2007). Mechanisms of strain localization and phase mixing operating in Ronda's mylonites were suggested to be bound to grain boundary sliding accommodated by dislocation creep (Précigout et al., 2007). Following Précigout et al. (2007), neighbour-switching during this process leads to scattering of orthopyroxene neoblasts within the olivine-rich matrix. This classical interpretation includes a general grain size reduction

in connection with the intensification of mixing towards the NW shear zone boundary which is thought to represent a strain localization with increasing stress (e.g., Garrido et al., 2011; Précigout et al., 2013). However, Ronda's peridotites in the complete shear zone area are extensively mixed. Thus, also the lesser deformed regions of the shear zone present in the distal mylonites and adjoining spinel tectonites are thoroughly mixed (e.g., Johanesen and Platt, 2015). Although mixing in the spinel tectonites was postulated to be melt derived (Johanesen et al., 2014; Soustelle et al., 2009), the mylonites were so far

considered to be either completely melt-free (Précigout et al., 2007; Soustelle et al., 2009) or melt-absent during the deformation (Johanesen and Platt, 2015). Johanesen and Platt (2015) challenged the classical view of Précigout et al. (2007) and proposed on base of a constant grain size of recrystallized olivine in the entire shear zone (mylonite + tectonites), an evolution of the shear zone at constant stress with increased strain rates in the NW.

With our detailed microstructural and geochemical study on samples taken across multiple transects across the mylonites and

into the tectonites (Fig. 1) we focus on the origin and evolution of the mylonitic unit. Analysis of the four major microstructural domains (mixed matrix, neoblast tails of clino-/orthopyroxene porphyroclast, olivine-rich matrix) decipher this evolution, which involves the interplay of melt, phase mixing and grain size reduction which in turn have a major impact on deformation and shear localization in the NW Ronda shear zone.



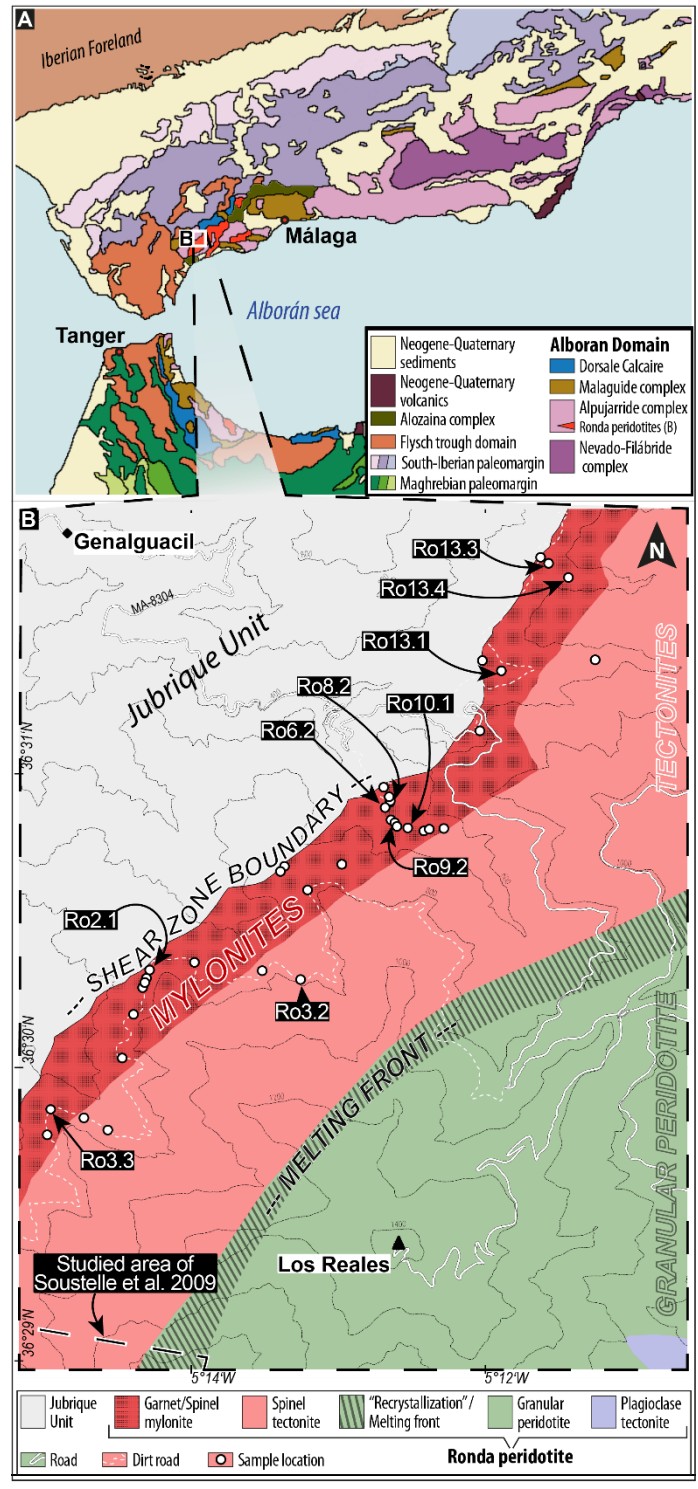



**Fig. 1. A: Geological overview of the Gibraltar Arc (Betic cordillera and Rif mountains) modified after Suades and Crespo-Blanc (2011). Ronda Peridotite indicated by white box. B: Close-up view of the NW Ronda peridotite modified after Précigout et al. (2013) & Van der Wal (1993) with sample locations.**

## 2 Geological setting

The Ronda peridotite, situated in southern Spain, is part of the Betic cordillera (Fig. 1). Together with the Rif mountains of
N Morocco it forms the Gibraltar arc (= Betic-Rif orogen), which surrounds the Alboran Sea. The Betic cordillera is subdivided into four tectonic domains: The external (1) Sub-Iberian and (2) Maghrebian domains formed by the paleomargins of Iberia and NE Africa, (3) the allochthonous Flysch trough unit comprising Mesozoic to Cenozoic sediments of the oceanic or continental Tethys, and (4) the internal, Alboran domain (Fig. 1A) (Booth-Rea et al., 2007). Separated by extensional shear zones and different metamorphic records, the Alboran domain is divided into three main tectonic
allochthons (Platt et al., 2006). In ascending order these are the Nevado-Filábride complex, the Alpujárride complex and the Maláguide complex. The lowermost Nevado-Filábride complex records a multistage metamorphic evolution including eclogite facies metamorphism followed by albite-epidote amphibole or greenschist facies overprints during decompression (Platt et al., 2006; Puga et al., 1999). The middle Alpujárride complex underwent HP-LT metamorphism overprinted by decompression and local heating (Balanyá et al., 1997; Platt et al., 2005). The uppermost Maláguide complex is
characterized by unmetamorphic rocks or by very low-grade metamorphism (Lonergan, 1993).

The peridotite bodies of Beni Bousera, Ojen, Carratraca and Ronda in sensu stricto are embedded as lenses in the upper Alpujárride complex (Platt et al., 2006). Superimposed on the Ronda peridotite, the Jubrique (or Casares) unit represents a highly attenuated crustal section of ≤ 5 km thickness (Fig. 1) (Barich et al., 2014; Obata, 1980). Near the contact to the Ronda peridotite, it displays MP-HT granulite facies kinzigites with melt inclusions (Balanyá et al., 1997; Barich et al.,
2014). With increasing distance to the peridotite, the metamorphic conditions decrease to LP-LT phyllites at the contact to the Maláguide complex (Balanyá et al., 1997). In the South and Southeast granitic rocks and migmatites of the Blanca unit underly the Ronda peridotite (Fig. 1). Partial melting and deformation of this unit at the contact to the peridotite have been attributed to the emplacement of the Ronda peridotite (Esteban et al., 2008). U-Pb SHRIMP dating of neo crystalline zircon rims from felsic and granitic dykes in this "dynamothermal aureole" point to an emplacement of the Ronda peridotite at 22.3
± 0.7 Ma (Esteban et al., 2011). Miocene, brittle, top-to-the-N extensional faulting led to the final emplacement of the Alpujárride complex with high cooling rates from 300-100 °C/Ma (Esteban et al., 2004; Platt et al., 2003; Rossetti et al., 2005).

### 2.1 The Ronda peridotite

With ca. 300 km$^2$ areal extent, the Ronda peridotite is the world's largest exposure of subcontinental mantle (Obata, 1980).
Its (micro) structural, petrological, and geochemical zoning led to its subdivision into four tectonometamorphic units (Fig. 1) (Précigout et al., 2013; Van Der Wal and Vissers, 1996, 1993). From NNW to SSE these are (1) a garnet/spinel-mylonite



unit, (2) a spinel-tectonite unit, (3) a coarse-grained granular-peridotite unit, and (4) a plagioclase-tectonite unit. Knowing that tectonites include mylonitic microstructures, we nevertheless adopt the established nomenclature and distinguish between "mylonites" and "tectonites". Thereby we hope to prevent confusion by multiple designations and enable a simple
reference to the literature.

The garnet/spinel mylonites, located along the contact to the Jubrique unit (~500 m thickness), are composed of fine-grained, porphyroclastic spl- and grt-bearing peridotites (lherzolites, harzburgites, dunites) (Van Der Wal and Vissers, 1993). They are strongly foliated (~N50° strike, 80° NW dip) with a nearly horizontal lineation (Précigout et al., 2013). Garnet-bearing pyroxenite layers are parallel to the foliation and predominantly stretched which leads in places up to their pinch-and-swell
type boudinage (Précigout et al., 2013; Van Der Wal and Vissers, 1993). Occasionally, the pyroxenite layers are also folded intrafolially with their fold axes oriented NE-SW slightly dipping towards the NE (Précigout et al., 2013). Graphitized diamonds in garnet-bearing pyroxenites and pre-deformational assemblages of olivine + pyroxenes + garnet found in pressure shadows indicate an origin of great depth (> 150 km) and a pre-mylonitic equilibration in the garnet stability field (1150 °C, 2.4-2.7 GPa (~100 km depth)) (Davies et al., 1993; Garrido et al., 2011). For mylonitic assemblages in the spinel
stability field, equilibration conditions of 800-900 °C and 1-2 GPa have been obtained by Johanesen et al. (2014), Garrido et al. (2011) and Van Der Wal and Vissers (1993).

The transition between grt/spl-mylonite and the spl-tectonite remains controversial. Contrary to cross-cutting contacts between mylonites and tectonites described by Van Der Wal and Vissers (1996), Précigout et al. (2007) postulated a continuous gradient from coarse grained tectonites (grain size of 250-450 µm) to fine-grained mylonites (150-220 µm).
Decreasing strain with increasing distance to the NW boundary of the mylonites is also indicated by decreased folding intensity and rotation of pyroxenite layers towards the SE (Précigout et al., 2013). However, Johanesen and Platt (2015) reported for both units (mylonites + tectonites) a consistent grain size of recrystallized olivine (~130 µm) and only an increase in the percentage of the recrystallized olivine grains towards the NW. As the main lithologies (harzburgites, lherzolites) and the foliation and lineation stay similar in tectonites and mylonites, tectonites were interpreted as the weaker
deformed counterpart of the mylonites (Van Der Wal and Vissers, 1993). Microstructural and geochemical data indicate additionally, that the tectonites were affected by melt impulses originating from the structurally lower, coarse granular peridotites (Johanesen et al., 2014; Soustelle et al., 2009).

Together with the grt/spl-mylonites, the spl-tectonites form the km-scale NW Ronda shear zone (Fig. 1). Its characteristics are the penetrative foliation with subhorizontal stretching lineation defined by cm-scale elongated orthopyroxenes and shear
criteria indicating sinistral kinematics and minor coaxial shortening (Balanyá et al., 1997; Précigout and Hirth, 2014; Van Der Wal and Vissers, 1996). Its boundaries are defined by the metasediments of the adjacent Jubrique unit in the NW and the underlaying coarse granular-peridotite unit in the SE (Fig. 1). Throughout the manuscript, the authors refer to the NW contact of the garnet-bearing mylonites which constitute the deformational center of the shear zone and the adjacent metapelites of the Jubrique unit as "shear zone boundary" (SZB). Sample locations are allocated with their distance [m] to



the SZB (Fig. 1). Even if not in its present appearance, the shear zone is considered to play a decisive role in the exhumation
of the peridotite massif (Johanesen et al., 2014; Précigout et al., 2013).

The coarse granular-peridotite unit is separated from the spl-tectonite zone by a "recrystallization"/ "coarsening" or
"melting" front (≤ 400 m) (Lenoir et al., 2001). Here, deformed grains annealed and coarsened, the foliation is lost and
garnet-pyroxenite layers are recrystallized as spl-websterites (Garrido and Bodinier, 1999). Lenoir et al. (2001) have shown

that the recrystallization front is the boundary/aureole of an area of partial melting (= coarse granular-peridotite unit) with
melt extraction < 5%. Secondary cpx, crystallized ahead of the front, proofed a re-fertilization (Soustelle et al., 2009). The
location of the front was shown to be strongly dependent on the peridotite solidus (1200° C) in regard to the temperature
gradient within the peridotite body (Lenoir et al., 2001). The coarse granular-peridotite unit itself is mainly composed of
unfoliated spinel harzburgite with minor lherzolite and dunite and various types of pyroxenites (Garrido and Bodinier, 1999).

The preservation of a strong olivine crystallographic preferred orientation (CPO) and folds of spl pyroxenites corroborates its
connection to the overlaying spl-tectonites (Vauchez and Garrido, 2001).

The youngest unit, overprinting the coarse granular-peridotite unit in the southeast, comprises the plagioclase tectonites
(Obata, 1980). Their equilibration at pressures of 0.8-0.9 GPa was placed in the context of the massif's exhumation in an
upper plate of a subduction zone (Van der Wal, 1993). It is composed of spl-free and spl-bearing plagioclase-peridotite

layers. The transition between both units records km-scale folding and shearing including the development of a new foliation
and the formation of mylonitic and ultramylonitic shear zones, which are tectonically assigned to the decompression of the
massif from spinel to plagioclase lherzolite facies prior to the emplacement into the crust (Hidas et al., 2013a).

## 3 Methods

Samples were cut perpendicular to the foliation and parallel to the stretching lineation (X-Z section). Thin sections of these

sections were polished to a thickness of ~30 µm. After optical analysis by polarization microscopy, electron backscatter
diffraction (EBSD) analysis combined with energy dispersive X-ray spectroscopy (EDX), and electron probe microanalysis
(EPMA) was performed on carbon coated thin sections. For EBSD and backscattered electron (BSE) analysis, thin sections
were polished beforehand with 0.03 µm colloidal silica.

Backscattered electron, EBSD and EDX analysis were conducted at the Institute for Geology and Mineralogy - University of

Cologne using a Zeiss Sigma 300-VP field emission scanning electron microscope (SEM) equipped with a NordlysNano
EBSD detector (Oxford Instruments). For a comprehensive overview, the entire thin sections were scanned in grids
simultaneously by BSE and EDX (O, Mg, Al, Si, Ca, Cr, Mn and Fe; Fig. 3). Having identified the microstructures of
interest, these were scanned simultaneously by EBSD, EDX, BSE and forescattered electrons (FSE). Measurement settings
were an acceleration voltage of 20 kV and a variable step size adapted according to grain sizes. Depending on the step size

and acquisition time, EBSD map sizes differ over a wide range. For data acquisition, the program AZtec 4.2 was used
(Oxford Instruments). The consistency of orientations between sample, measurement and post-processing reference frame



was ensured by the measurement of a quartz standard. It consists of four synthetic quartz crystals embedded in epoxy. The known positions of the quartz single crystals in the standard combined with their known individual orientation enables the operator to identify possible rotations (spatially or crystallographic) of the data during acquisition and processing. Kilian et al. (2016) have shown that such rotations occur often due to unknown orientation in sample material and mistranslations between different processing platforms. Obtained EBSD data was as first step cleaned by deleting "wild spikes" and filling not indexed points with the average orientation of 6 or more neighbour orientations of the same phase (HKL Channel 5 software - Oxford Instruments). Additionally, the EBSD data were corrected for systematic mis-indexing of olivine due to similar diffraction patterns for orientations rotated 60° around [100]. Secondly, the cleaned data were imported into the MTEX 5.7 MATLAB extension (e.g., Bachmann et al., 2010). All following data processing and analysis were conducted using MTEX (http://mtex-toolbox.github.io/). Orientations of indexed points with high mean angular deviations (MAD > 1) were filled by the mean orientation of the neighbouring points. After grain calculation (grain internal misorientation < 15°) grain size specific and inclusion deletion and/or filling was carried out individually for each map. Incomplete grains at the borders of the mapped areas and badly indexed grains were excluded from further analysis. During the cleaning and reconstruction, the results were checked against backscattered/forescattered, band contrast and microscope images. The EBSD phase assignment was checked by simultaneously obtained EDX maps and/or EDX point measurements. The cleaned EBSD maps were thereupon analyzed for grain and phase properties, boundary properties and orientation properties. Analyzed grain properties are phase abundances by covering area percentage, grain amount, grain size by the equivalent circular diameter (ECD), grain shape by aspect ratio, shape factor and shape preferred orientation (SPO). Phase abundances given by "%" in figures and in the entire manuscript are referring to area percentages. Boundary properties are grain (phase A- phase A) and phase (phase A- phase B) boundary percentages calculated by phase specific boundary length. The ratio of total grain to total phase boundary length gives the "mixing intensity" of a microstructure. Orientation properties include phase-specific crystallographic orientations illustrated by orientation or orientation density function (ODF) stereo plots. Its strength is calculated by the M-index, the J-index and the maximum mrd value (multiple of random distribution). Both, J-index (Bunge, 1982) and M-index (Skemer et al., 2005), express the strength of a given ODF. For a detailed evaluation of both see Skemer et al. (2005). All pole figures are equal-area lower-hemisphere plots. ODFs are only displayed for a minimum number of 100 grains per phase. Otherwise, single orientations are plotted in the stereoplot as dots. ODFs were calculated with grain mean orientations and a consistent halfwidth of 15°. To facilitate the comparison between ODF plots, the color-coding range is fixed from blue (mrd= 0) to red (mrd= 3). Higher mrd values are accordingly also colored red.

Microprobe measurements of olivine, clinopyroxene, orthopyroxene, spinel and amphibole were conducted at the Institute of Geosciences - Goethe University Frankfurt a.M. using a field emission JEOL JXA-8530F Plus microprobe equipped with 5 wavelength-dispersive spectrometers. Measuring settings were 15 kV acceleration voltage and 20 nA beam current for 20 s (Al, Cr, Ca, Na, Mn, Fe and Ni), 30 s (P, K and Ti) or 40 s (Mg and Si) peak and 20 s for background measurement (settings and detection limits in S1). The same measurement settings and standards were used for the analysis of all phases. The spot-size was adjusted to the grain size with minimum sizes of 1 µm for small neoblasts and maximum 4 µm for porphyroclasts.



References to supplementary data are given in the text by an "S" combined with the number of the appendix (e.g., S3 for supplementary CPO data).

## 4 Results

Samples were taken over a range from 39 to 703 m distance to the shear zone boundary (Fig. 1). Referring to the established
subdivision of the Ronda peridotite developed by Précigout et al. (2013) and Van Der Wal and Bodinier (1996) our samples are dominantly taken from the garnet-spinel mylonites and in greater distance from the shear zone boundary (SZB) from the spinel tectonites (Fig. 1). Both units are composed of lherzolite or harzburgite with minor dunitic lenses. The samples have a highly variable degree of serpentinization. Consistent with previous studies, the foliation is mostly oriented parallel to the SZB and steeply dipping (65-85°) (Précigout et al., 2013; Soustelle et al., 2009; Van Der Wal and Vissers, 1996, 1993;
Vauchez and Garrido, 2001). Towards the contact, the foliation intensifies. In general, the samples display a NE-SW trending foliation dipping towards NW with subhorizontal to shallowly SW dipping (<20°) stretching lineation. The latter is defined by elongated orthopyroxene singe crystals, neoblast tails of pyroxene porphyroclasts and olivine-rich lenses stretched in the foliation plain. Towards the contact to the bordering Jubrique metasediments, macroscopic evidence for increasing strain is an increase of porphyroclast elongation (Fig. 2) and an increase of the mylonitic matrix (Fig. 3). The
elongation is mostly visible by the lengthening of orthopyroxene single crystals, which expand to aspect ratios > 10:1 and length of ~5 cm (Fig. 2). The increasing strain is additionally indicated by increasing deformation of partly garnet-bearing pyroxenites, by pinch-and-swell structures (Fig. 3D-i), and by boudinage described in detail by Précigout et al. (2013).

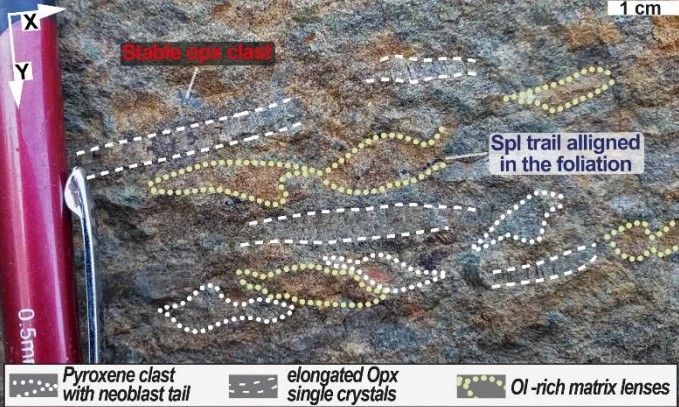

**Fig. 2. Field photograph of a mylonitic sample (Ro3.3, for location see Fig. 1) with strongly elongated orthopyroxene single crystals**
**(XY section).**

### 4.1 Microstructures

Figure 3 gives a microstructural overview of representative thin sections with increasing distance to the SZB. Due to serpentinization and to facilitate phase identification, BSE and Ca-EDX overview scans are shown instead of microscopic



images. The proportion of neoblasts in the matrix increases towards the shear zone boundary (Fig. 3). Simultaneously, the
abundance of porphyroclasts decreases. Nevertheless, deformation features like a clear foliation with marked elongation of
porphyroclasts and recrystallized olivine dominated matrix are present in all mylonitic samples (39-502 m distance SZB, Fig.
3). Only the outermost, tectonic sample (703 m distance SZB) lacks these features and shows a relatively undeformed fabric
(Fig. 3F). Even though deformation was not as localized in this sample as in those situated closer to the SZB, interstitial
pyroxenes along olivine grain boundaries are present (Fig. 3F-ii). Furthermore, layers consisting of a pyroxene and spinel
assemblage crosscut the tectonic peridotite. Approaching the SZB, pyroxene porphyroclasts show neoblast tails, which
stretch out in the foliation. Simultaneously, pyroxenite layers turn parallel to the foliation and flatten till they disintegrate
(Fig. 3D-ii).

With the focus on phase mixing as well as on reaction and recrystallization processes, we further investigated structures on
the micro-scale rather than on thin-section or larger scale. By microscopic analysis and the BSE/EDX element thin section
overviews (Fig. 3) four major microstructural domains were identified: (1) Olivine-rich matrix, (2) mixed matrix, (3)
clinopyroxene neoblast tails, and (4) orthopyroxene neoblast tails. In the following, microstructural characteristics of these
four domains are presented. Additional to these four main domains, amphibole-clinopyroxenite veins investigated in three
thin sections will be shortly addressed. For reasons of length and clarity, only a selection of microstructures is shown in
figures, providing an overview of each domain and their change depending on the distance to the SZB. These figures include
ODFs referring to the depicted microstructure. Graphs of the complete microstructural data are presented in Fig. 4. Average
aspect ratios and average grain size are included only if enough grains were present for a valid statistical analysis (n> 20).
The results presented below include all analyzed microstructures of a given microstructural domain. The complete data is
attached as supplementary data (S2). Garnet, even if nominally present in a few maps (39 of 41563 analyzed grains), is
excluded from further analysis because of its small abundance (< 0.1 % for all microstructural domains) and its susceptibility
for mis-indexing with opx especially for small grains. Coarse grained garnet (ECD > 100 µm) was not present in the studied
microstructures.









**Fig. 3. Electron backscatter (left) and Calcium energy-dispersive X-ray spectroscopy (right) scans of the same thin sections ordered with increasing distance to the shear zone boundary (black numbers on white background). Locations of analyzed**
**example microstructures and their figure # (5-9) are indicated. Dominant microstructural domains of the NW Ronda shear zone are marked: (1) strongly serpentinized olivine rich matrix lenses, (2) mixed matrix with interstitial pyroxenes, (3) ortho- and (4) clinopyroxene porphyroclasts with neoblast tails. Note the presence of stable pyroxenes and elongated orthopyroxene. The presence of interstitial cpx (bright colours in EDX images) is indicative for the mixed matrix. D-i includes a close up on kelyphitic assemblage.**

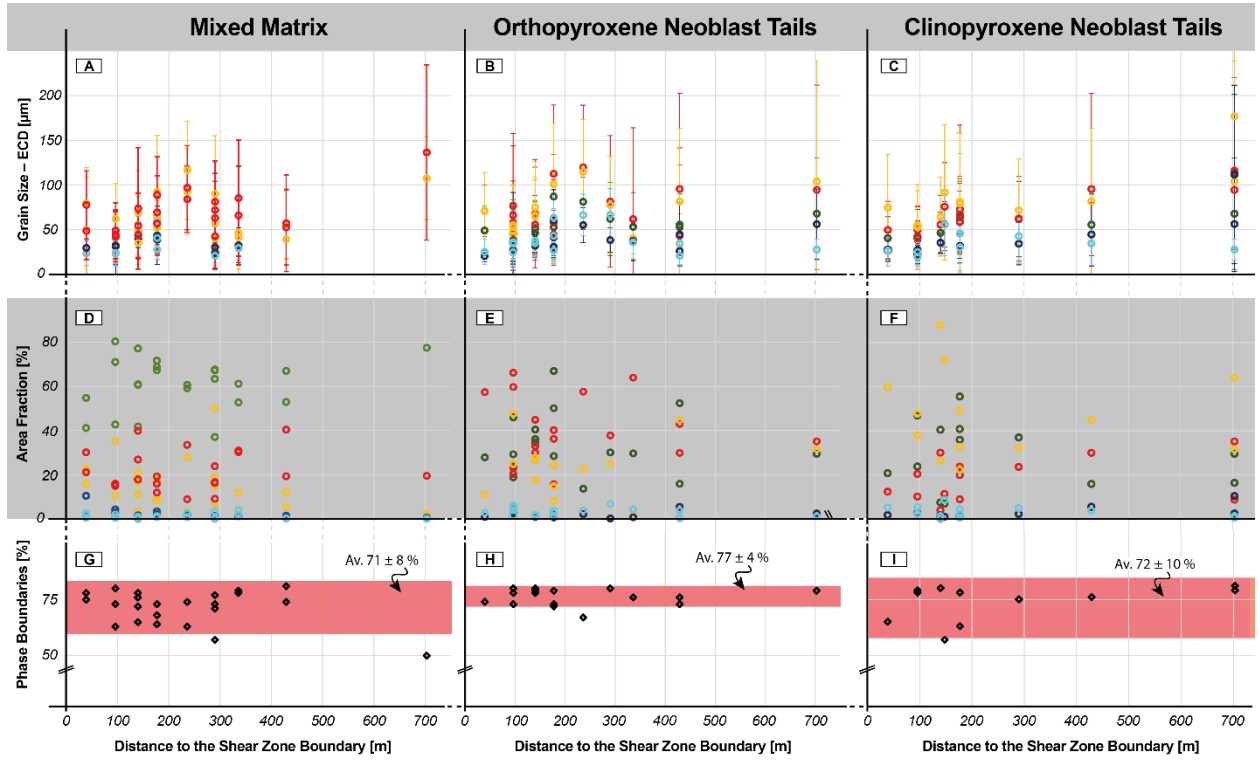


**Fig. 4. Data overview of the major microstructural domains (mixed matrix, orthopyroxene and clinopyroxene porphyroclast neoblast tails) plotted against the distance to the SZB. The olivine-rich matrix was excluded due to its small data base. Each data point represents the average grain size (A,B,C), area fraction (D,E,F) or phase boundary percentage of the total boundary length (G,H,I) of one EBSD map of an analyzed microstructure (e.g., Figs. 5-8). The complete microstructural data is attached in**
**supplementary data 2.**

### 4.1.1 Matrix domains

The overall olivine dominated matrix forms a major part of all analyzed samples (Fig. 3). Due to the presence of interstitial cpx (high Ca counts in Fig. 3) and opx, most of this matrix is mixed with only parts remaining almost monomineralic, olivine rich. Even though, lenses of olivine-rich matrix domains are present in all samples (Fig. 2). However, the
differentiation between mixed domains and olivine dominated domains becomes increasingly difficult with decreasing distance to the SZB. In both matrix domains olivine grains are cut by subvertical or subhorizontal serpentine veins. With increased degree of serpentinization, olivine grain boundaries become increasingly lobate and originally coherent grains are




separated into smaller fragments. Coherent crystallographic orientations with bended lattices over span multiple fragments, which were identified as single grains by the EBSD analysis (Fig. 5). Due to this discrepancy between calculated fragment

grains and pristine, pre-serpentinization grains, the grain properties (size, aspect ratio etc.) of the calculated grains of olivine are excluded from further analysis.

### 4.1.1.1 Olivine-rich matrix

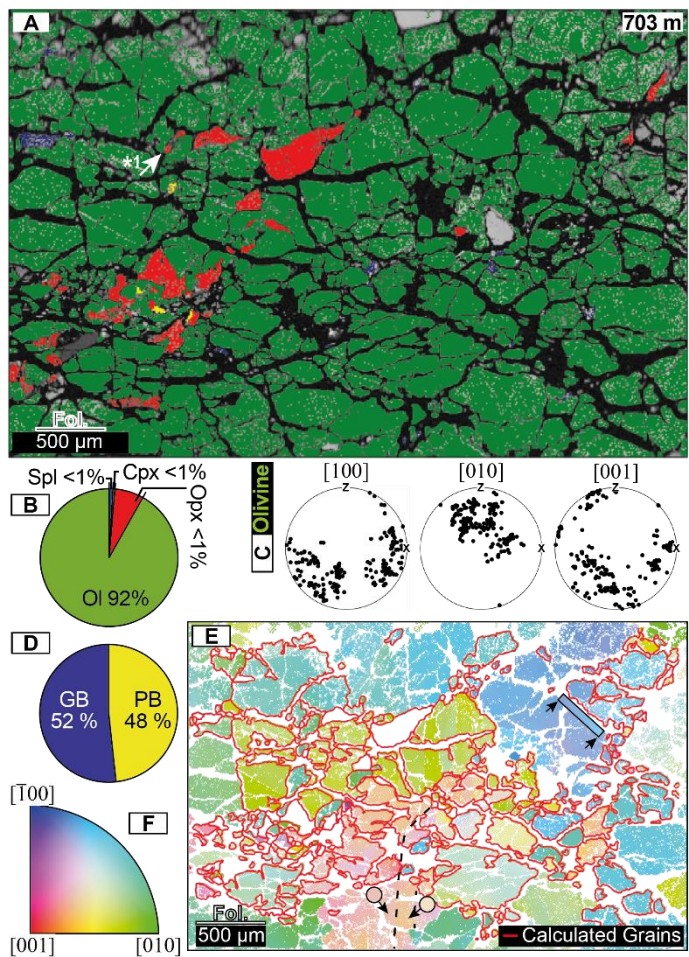

**Fig. 5. A: Example microstructure of olivine-rich matrix in sample from 703 m distance to the SZB. Area percentages (B),**
**orientations of calculated olivine grains (C), and phase and grain boundary percentages (D) are given. The orientation map of olivine (E) shows the areas of original grains identifiable by similar orientations. Examples of intracrystalline deformation by bending of the crystal lattice (box) and subgrain boundary (dashed line) covering and crossing multiple calculated fragmental grains (red outlines) are annotated. The orientation colour key is given in F.**

Due to the increased serpentinization in olivine-rich domains and due to the pervasive occurrence of pyroxenes as interstitial
grains and on olivine grain boundaries only two microstructures of the olivine rich matrix type could be analyzed (e.g., Fig. 5). They consist of on average 93 % olivine (range 92-84), 6 % opx (range 6-7) and minor spinel (1 %) and cpx (< 1 %). The



crystallographic orientations of the olivines are predominantly parallel to each other (S3). Areas of parallel orientations include multiple of calculated olivine grains (Fig. 5). In cases, bended lattices or subgrain boundaries can be traced over the boundary of calculated grains (Fig. 5 – box and dashed line). Even if statistically not verifiable, the size of these olivine areas which, in the opinion of the authors, represent the original grains prior to crosscutting serpentinization, was estimated to be several cm (Fig. 5). Cracks within those original grains and along their boundaries are filled with serpentine (Fig. 5). Orthopyroxenes within the olivine-dominated matrix are oriented with their [001] axis perpendicular to the foliation plane (S3).





## 4.1.1.2 Mixed matrix

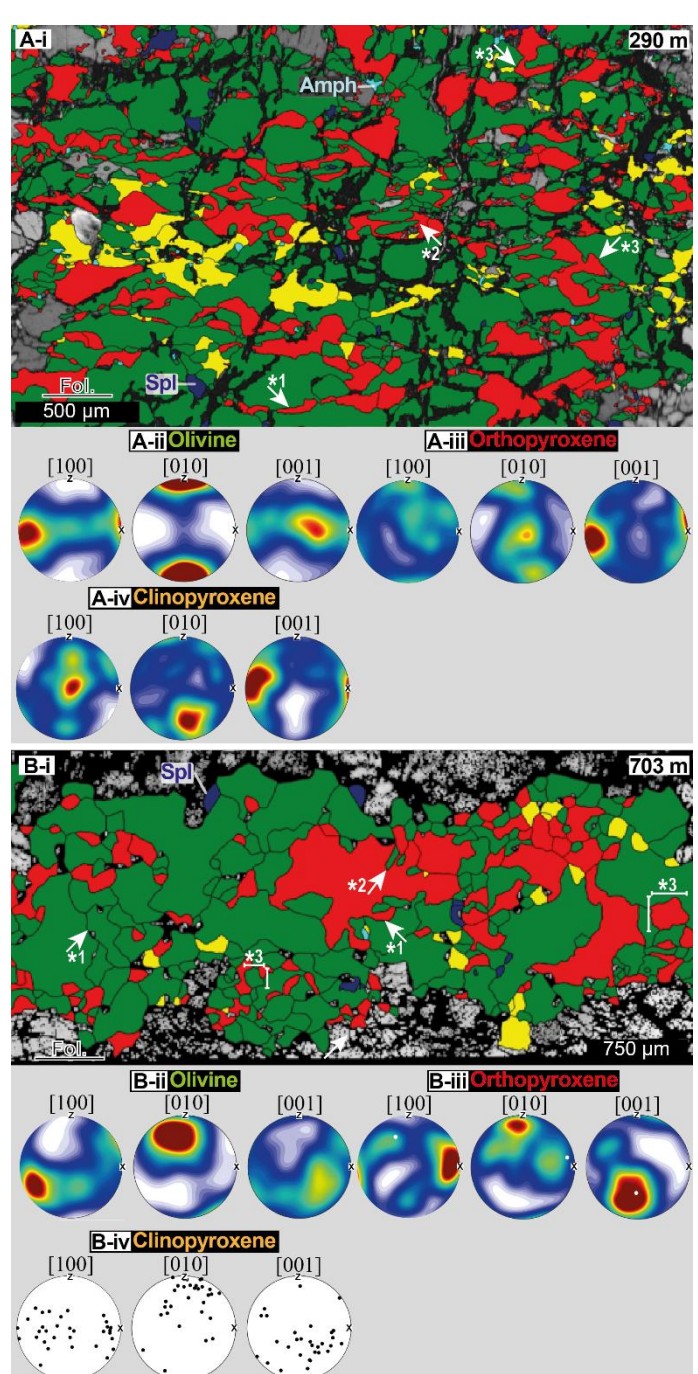


**Fig. 6. Example microstructures of the mixed matrix in 290 m (A-i) and 703 m (B-i) distance to the SZB. A: Mixed matrix with wedge/film-shaped orthopyroxene (\*1) in between coarser olivine, indentations (\*2) and highly irregular phase boundaries (\*3). CPOs of olivine (A-ii), orthopyroxene (A-iii) and clinopyroxene (A-iv). B: Mixed matrix of a tectonite sample with highly lobate**



grain and phase boundaries (*1), indentations (*2) and equi-axial grain shape (*3). CPOs of olivine (B-ii) and orthopyroxene (B-iii)
and pole figure of clinopyroxene orientations (A-iv).

The microstructures of 23 mixed-matrix domains were analyzed either as subsets of EBSD maps covering pyroxene
porphyroclast tail assemblages with adjacent matrix or as individual map. The subsets only consist of the mixed matrix
surrounding the porphyroclast-tail assemblage. As mixed matrix microstructures are present in the entire transect (703 to 29
m distance to SZB), they are present in both mylonitic and tectonite samples. Characteristic for the mixed matrix are small,
interstitial and irregular/highly lobate pyroxene grains in between coarser grained olivine (Fig. 6). The grain shape of
pyroxenes and olivine varies between rather equi axial in tectonic samples (Fig. 6B-i) to elongated olivines and film/wedge-
shaped pyroxenes in mylonitic samples (Fig. 6A-i). The long side of film-like pyroxenes is mostly parallel to the foliation.
Pyroxenes have in general a highly irregular grain shape and show in places intergrow patterns with olivine (Fig. 6A-i *2).
The mineralogical assemblage of the mixed matrix is in general constant for the entire data set and consists of olivine (av. 61
%, range 37-80 %), orthopyroxene (av. 22 %, range 9-40 %), clinopyroxene (av. 14 %, range 2-50 %) and spinel (av. 2 %,
range 0-10 %) with occasional amphibole (av. 1 %, range 0-4 %) (Fig. 4D). Spinel is present as interstitial grains. Phase
abundances vary depending on the microstructural setting within one sample but do not significantly change over the
distance to the SZB (Fig. 4D). Similar to the olivine-rich matrix, former coarse grained olivines are cut by serpentine veins.
Only in phase mixtures olivine grains are less effected by the serpentinization. Average grain sizes (ECD) are 67 µm for opx
(range 42-136), 64 µm for cpx (range 35-117), 41 µm for spinel (range 24-90) and 40 µm for amphibole (range 21-60) (Fig.
4A, S2). Almost over the entire mylonitic transect (39-429 distance to the SZB) grain sizes of the mixed matrix are similar
within uncertainty (1σ, Fig. 4A). Only around ~250 m distance to the SZB, the grain size of both pyroxenes shows an
excursion towards coarser sizes. Mixed matrix pyroxenes in the tectonite regime (Fig. 6B-i) have coarser grain sizes (Fig.
4A). Average aspect ratios are 1.9 ±.2 for opx, 1.9 ±.3 for cpx, 1.8 ±.1 for spinel and 1.8 ±.2 for amphibole (S2). In contrast
to the grain sizes, the average aspect ratios remain constant over mylonites and tectonites (S2). 71 ±8 % of the total boundary
length are on average phase boundaries (29 % grain boundaries). Apart from one outlier, all mylonitic mixed matrix domains
share this distribution. In the tectonite, phase boundaries only form 50 % of the total boundary length. On average 40 % of
the total boundary length are olivine-opx boundaries with 48 % of the entire olivine boundaries and 79 % of all opx
boundaries being olivine-opx boundaries. Despite the lower abundance of cpx, amphibole forms more phase boundaries with
cpx (29 %) than with opx (16 %).

Olivine CPOs are moderate (av. max mrd 10, av. M-index 0.17). Overall, the A-type olivine CPO is dominant (18 of 23
mixed matrix microstructures). However, transitions to the AG-type by [100] and [001] forming girdles in the foliation plane
are present with variable strength (clear AG-type CPO n=3). Clear B-types are also present in the two samples situated
closest to the SZB. Orthopyroxene CPOs are with an average maximum mrd of 8 and an average M-index of 0.06 the
weakest opx CPOs of all investigated domains. In most cases, orthopyroxene's [001] axes are parallel to the lineation. The
CPO of opx neoblasts is in places affected by the orientation of larger opx grains within the mixed matrix (Fig. 6B-iii).
Clinopyroxene CPOs are weak with an average M-index of 0.09 (av. max mrd 18). In most cases, both pyroxenes are



oriented parallel to each other and show similar intensities (mrd, M-index) for a given microstructure. In some cases, maxima of pyroxene [100] and [010] orientations are flipped in the sense that clinopyroxene [100] maxima are parallel to

orthopyroxene [010] maxima, and cpx [010] display orientations similar to opx [100] (e.g., Fig. 6A-iii/iv). Only in four mixed matrix microstructures enough amphibole grains are present to determine a CPO (S3). In general, amphibole orientations are parallel to the present pyroxene and its [001] axes are aligned parallel to the lineation.

### 4.1.2 Porphyroclast tails

Porphyroclasts are present in all mylonitic samples (Fig. 3). In the tectonite sample, the small difference between grain sizes

of matrix and porphyroclasts does not allow a clear differentiation between both (Fig. 3F). Here, pyroxene is either present in layers consisting of both pyroxenes, spinel and minor olivine (Fig. 8B-i), as clasts, or (Fig. 3F-i/ii) as interstitial pyroxenes along grain boundaries of olivine clasts (Fig. 6B-i). The pyroxene porphyroclasts in tectonites are predominantly orthopyroxenes. In mylonitic samples, the contrast between porphyroclasts and matrix is marked by their strongly differing grain sizes (Fig. 3A-E). Most porphyroclasts are pyroxenes. Often assemblages of intergrown pyroxenes ± spinel form

porphyroclastic assemblages (Fig. 3D-i). These assemblages are predominantly present in deformed pyroxenitic layers or in areas with an increased pyroxene proportion (Fig. 3C-i lower image half). Occasionally, also garnet surrounded by kelyphitic rims or coarse spinel grains (Fig. 2E-i) form porphyroclasts. As both pyroxenes are present as (porphyro)clasts in tectonites and mylonites, their microstructures were analyzed from 29 to 703 m distance to the SZB. In mylonitic samples, both pyroxenes show the formation of neoblast tails (Fig. 3A-E). Common characteristics of pyroxene neoblast tails are a mixed

phase assemblage of both pyroxenes, olivine, amphibole and spinel. The affiliation of neoblast tails to the parent porphyroclasts is given by the contrast in phase composition and neoblast grain shape between tail and surrounding matrix as well as by neoblast indentations into the parent porphyroclast (Figs. 7,8).




## 4.1.2.1 Orthopyroxene porphyroclasts

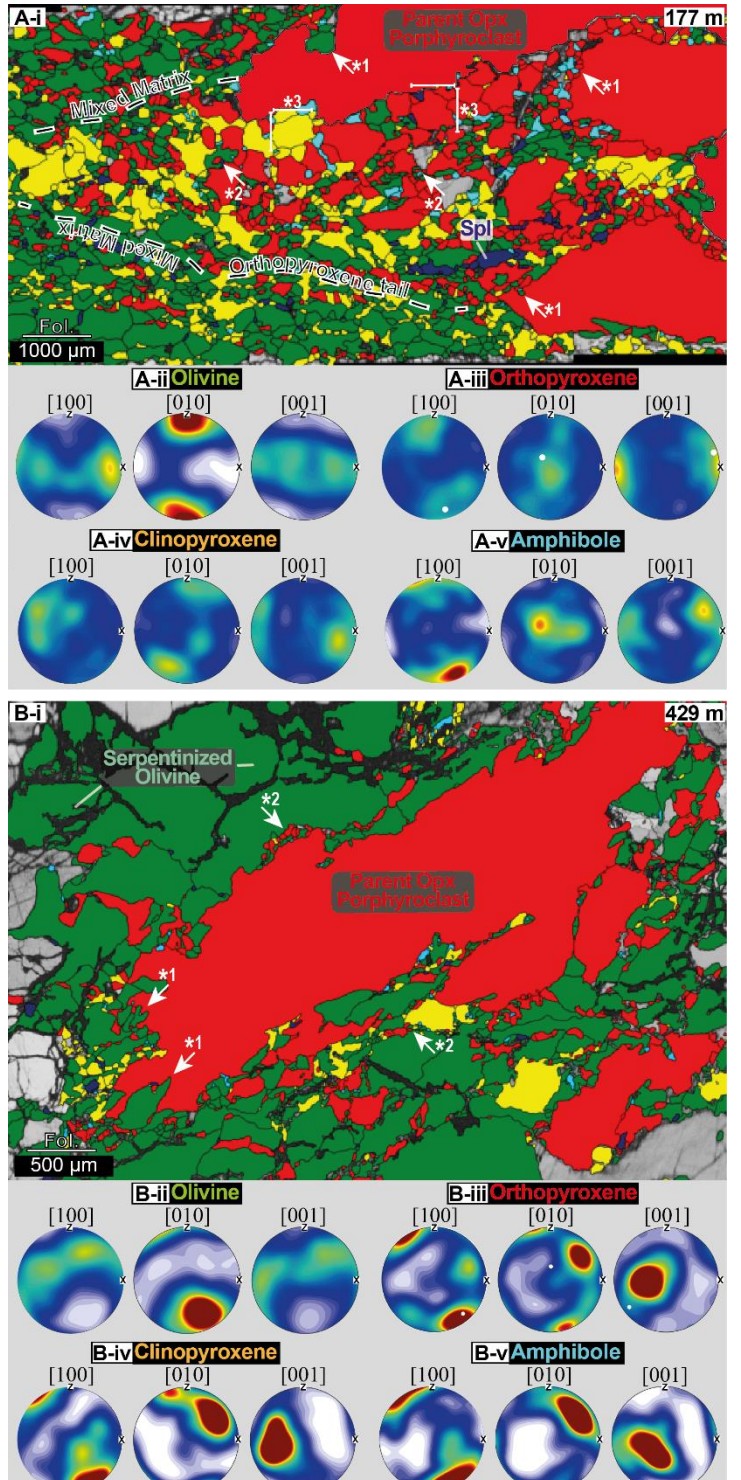





**Fig. 7. Example microstructures of orthopyroxene porphyroclasts with neoblasts in 177 m (A-i) and 429 m (B-i) distance to the SZB. A: Neoblast tail of opx and subordinate cpx porphyroclast assemblage. Annotated are neoblast indentations in parent porphyroclasts (\*1), interstitial amph and spl (\*2) and equi-granular grain shape (\*3). Note the difference in phase composition and abundances, grain size and shape between neoblast tail and surrounding mixed matrix. CPOs of all present phases are given in A ii-v with white dots in A-iii indicating the parent clast orientation. B: Orthopyroxene porphyroclast with neoblast**
**indentations (\*1) and fine-grained mixed neoblast assemblages at its boundary (\*2). Note the presence of fine-grained mixed neoblast along grain boundaries of the surrounding coarse olivine. CPOs of all present phases are given in B ii-v with white dots in B-iii indicating the parent clast orientation.**

Orthopyroxene clasts (tectonites) and porphyroclasts (mylonites) are present in all samples. Their shape is variable (Fig. 3).

However, towards the SZB highly elongated porphyroclasts (aspect ratios > 1:10) become more abundant. Neoblast

formation around opx porphyroclasts is present in all mylonitic samples. Common characteristics of neoblasts are low

internal deformation, equi-axial grain shape and often irregular boundaries (Fig. 7). In the distal part of the mylonite unit, the

formation of neoblasts is weaker and rather arranged in diffuse patches around the porphyroclast (Fig. 7B-i). Neoblast

assemblages are present at the parent clast grain boundary and extend along grain boundaries into the surrounding coarse-

grained olivines (Fig. 7B-i). With decreasing distance to the SZB, opx porphyroclast neoblast assemblages become more

abundant and form tails within the foliation (Fig. 7A-i). The mineralogical assemblage of these domains consists of

orthopyroxene (av. 41 %, range 16-66), olivine (av. 34 %, range 14-67), clinopyroxene (av. 20 %, range 1-48), amphibole

(av. 3 %, range 1-7) and spinel (av. 2 %, range 1-6) (Fig. 4E). Spinel and especially amphibole form mostly interstitial

grains. There are no clear trends in the phase assemblage related to the distance to the SZB (Fig. 4E). Amphibole and spinel

are constantly present as secondary phases with standard deviations of ±2 % (amph) and ±1 % (spl) (Fig. 4E). For olivine

and both pyroxenes, phase abundances in opx neoblast tails can vary in a single thin section in the same magnitude as over

the entire shear zone transect. Average grain sizes are 69 µm for opx (range 41-120), 54 µm (range 38-81) for olivine, 66 µm

for cpx (range 34-115), 37 µm for amphibole (range 21-67) and 36 µm for spinel (range 20-56) (Fig. 4B). Apart from one

excursion at around 250 m distance to the SZB, the grain sizes are largely constant throughout the entire transect (Fig. 4B).

For a given opx tail, grains of both pyroxenes are mostly similar sized (± 10 µm). Amphibole and spinel have similar, small

grain sizes with ECDs in general half the size of pyroxene neoblasts. Average aspect ratios are with 1.8 for opx, 1.9 for

olivine, 1.8 for cpx, 1.8 for amphibole and 1.9 in general lower than in matrix domains (S2). In contrast to the grain size,

aspect ratios remain constant in all mylonitic samples. In the tectonite, aspect ratios tend to be higher (S2). Phase boundaries

clearly dominate (77 ±4 %) over grain boundaries (Fig. 4H). Apart from one outlier, these high phase boundary percentages

are present over the entire shear zone (Fig. 4H). Although opx is mostly the predominant phase, olivine forms, on average,

most of the phase boundaries (S2). Olivine neoblast CPOs are the weakest for all microstructural domains (av. max mrd 9,

av. M-index 0.14). Dominant is the AG-type CPO with girdle distributions of [100] and [001] in the foliation plane (n=11;

S3). Transitions to A- or B-type CPOs are formed by point maxima in these girdles (Fig. 7A-ii). Two clear A- and B-type

CPOs are present for olivine neoblasts in opx tails (S3). Orthopyroxene neoblasts have the strongest opx CPOs of all

microstructural domains (av. max mrd 12, av. M-index 0.09). For almost all orthopyroxene porphyroclast-neoblast

assemblages, opx neoblast CPOs are strongly dependent on the parent clast orientation (e.g., Fig. 7A/B-iii). This



porphyroclast dependence is present in both strong and weak CPOs of orthopyroxene neoblasts. The common orthopyroxene CPO is [001] parallel to the lineation. The [100] and [010] maxima do not show such a clear trend. Clino- and orthopyroxene CPOs are always for [001] and predominantly for [100] and [001] parallel to each other (e.g., Fig. 7B-iii/iv). With an average maximum mrd of 19 and an average M-index of 0.12 cpx neoblasts in opx porphyroclast tails form the strongest clinopyroxene CPOs of all microstructural domains. For all orthopyroxene tails, amphibole CPOs are related to the orthopyroxene neoblast CPOs and thereby also parallel to the parent clast orientation (e.g., Fig. 7A/B-v).



### 4.1.2.2 Clinopyroxene porphyroclasts

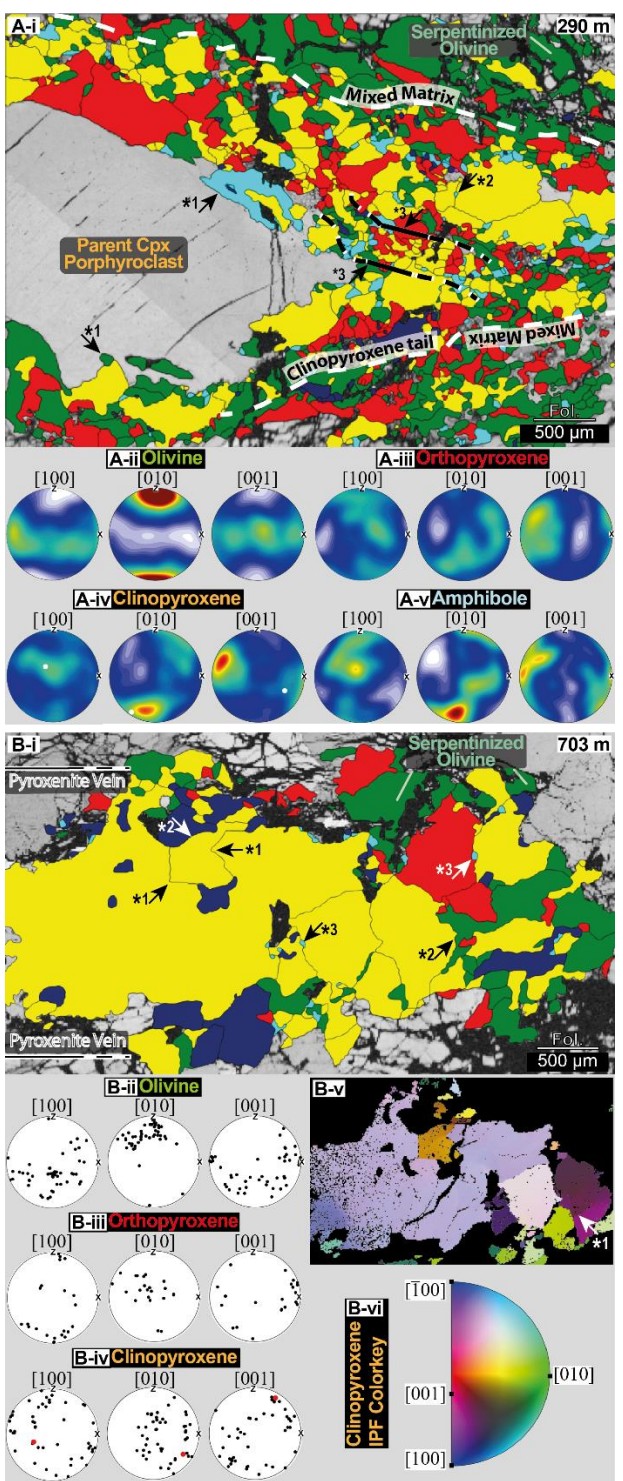





**Fig. 8. Example microstructures of clinopyroxene porphyroclast with neoblast tail in 290 m (A-i) and cpx-dominated pyroxenite layer at 703 m (B-i) distance to the SZB. A: Clinopyroxene porphyroclast neoblast tail embedded in the mixed matrix with amphibole/ol indentations (\*1) and interstitial amph (\*2). Within the tail a band of fine grained neoblasts (\*3) is present. CPOs of all phases are given in A ii-v. B: Pyroxenite layer with straight grain (\*1) and lobate/irregular phase boundaries (\*2) and interstitial amph and spl (\*3). Pole figures of ol (B-ii), opx (B-iii) and cpx (B-iv) orientations are given. The cpx orientation map (B-v; colour key in B-vi) shows grain-internal deformation and subgrain boundaries (\*1).**

In the tectonite, isolated clasts of clinopyroxene are less frequent than those of orthopyroxene (Fig. 3F). Here, beside the small, interstitial cpx grains mentioned in the mixed matrix section, coarser clinopyroxene grains are predominantly present in pyroxenite layers consisting of intergrown pyroxenes, spinel and olivine (Fig. 8B-i). Phase boundaries in these are irregular, whereas grain boundaries tend to be straight and angular (Fig. 8B-I \*1). A differentiation between parent clast and neoblasts is not possible. In mylonitic samples, clinopyroxene porphyroclasts, either present isolated (Fig. 3B-ii) or in deformed assemblages of the above-described layers (Fig. 3D-ii), form tails of neoblasts, which are sweeping into the foliation. Compared to opx neoblast tails, those of clinopyroxene porphyroclasts are more pronounced both in frequency and in tail length (Fig. 3). Additionally, contrary to opx neoblast tails, which formation/frequency seems to depend on the proximity to the SZB, neoblast tails of cpx porphyroclast are also present in distal mylonitic samples (Fig. 3E-ii). Neoblast tails of cpx porphyroclasts consist of 48 % clinopyroxene (range 22-88), 27 % olivine (range 7-56), 19 % orthopyroxene (range 4-40), 3 % amphibole (range 0-9) and 2 % spinel (range 1-10). For the major components (cpx, opx, ol) no change in phase abundances is present over the transect (Fig. 4F). For the minor phases of amphibole and spinel it seems that in distal parts of the mylonites and in tectonites spinel is the prevailing secondary phase, whereas closer to the SZB amphibole is more abundant (Fig. 4F). In most microstructures, spinel and amphibole occur separated from each other. Average grain sizes are 88 µm for cpx (range 53-177), 57 µm for olivine (range 29-115), 70 µm for opx (range 40-116), 38 µm for amphibole (range 20-63) and 44 µm for spinel (range 23-112). The distribution of grain sizes is divided into coarse areas of primary phases (pyroxenes and olivine) and fine-grained areas of thoroughly mixed secondary (amphibole or spinel) and primary phases (Fig. 8A-I \*3). Primary phase grain sizes are relatively constant over the first 300 m distance to the SZB (Fig. 4C). In the distal mylonitic part and in the tectonic regime, their grain size increases. Amphibole and spinel average grain sizes are about half of the size of primary phases (Fig. 4C). Their grain sizes tend to be constant over the entire transect. For both, primary and secondary phases, a slight excursion towards bigger grain sizes around ~280 m distance to the SZB is present. Average aspect ratios are with 1.8 for cpx, 1.9 for olivine, 1.9 for opx, 1.9 for amphibole and 1.9 in general lower than in matrix domains (S2). In contrast to the grain size, aspect ratios are more constant over the entire transect (S2). Phase boundaries form on average 72 % (±10 %) of the total boundary length (Fig. 3I). This distribution is in general constant over the entire transect, independent of mylonitic or tectonic unit. Amphibole is mostly affiliated to clinopyroxene (S2).

Olivine neoblasts CPOs in tails of clinopyroxene porphyroclasts are variable. Beside the most present A- and B-type (each n=4) transitions to the AG- type with point maxima in [100] and [010] girdles, pure AG-types and one clear E-type are present (S3). Their strength is moderate to strong (av. max mrd 12, av. M-index 0.15). Clinopyroxene neoblast CPOs are weak (av. max mrd 15, av. M-index 0.09). In most cases, the parent clinopyroxene porphyroclasts have an imprint on the



440 neoblast orientation (e.g., Fig. 8A-iv). However, compared to orthopyroxene, clinopyroxene maxima are often less pronounced and blurred and therefore more variable from their parent clast orientation. The [001] axes are largely parallel to the lineation. Occasionally (n=2), [100] maxima are oriented parallel to the lineation. If present, orthopyroxene neoblasts are parallel to clinopyroxenes with their [001] and show occasionally 90° rotations for [100] and [010]. Amphibole neoblasts are mostly oriented parallel to the pyroxenes (e.g., Fig. 8A-v).

### 4.1.3 Clinopyroxene-amphibole veins

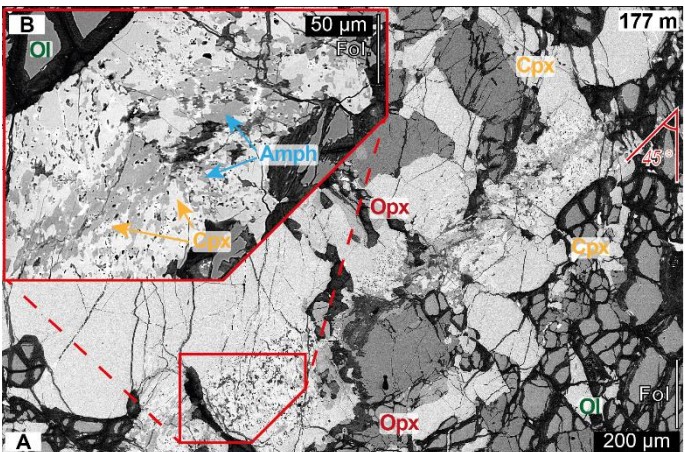

**Fig. 9. A: Clinopyroxene amphibole vein crosscutting the mixed matrix and a clinopyroxene porphyroclast. Note the ~45° angle between foliation and vein orientation. B: Close-up view of vein showing zoning in cpx-rich rim and amph-rich center.**

In three samples, veins consisting of fine grained (ECD < 10 µm) cpx and amphibole were analyzed. These veins crosscut 450 porphyroclasts, tails and the mixed matrix with an orientation of ~45° to the foliation (Fig. 9). In olivine-rich domains no such veins were found. The veins are best visible when crosscutting cpx porphyroclasts or pyroxenite layers (Fig. 9). Crosscutting previous structures oblique to the foliation, these veins are attributed to the late evolution of the Ronda peridotite. Because of the focus of this research on the mylonite formation a detailed microstructural investigation was omitted. However, a short revision is given in the geochemistry and discussion chapters.

### 4.2 Mineral chemistry

The major-element composition of both pyroxenes (opx and cpx), olivine, amphibole and spinel were determined from three samples with different distances to the SZB (96 m, 177 m, 290 m). Apart from the olivine-rich matrix, all microstructural domains (cpx/opx neoblast tails, mixed matrix) were analyzed for each sample, if present. Neoblast tail measurements include the analysis of the parent pyroxene porphyroclast. There is a general trend for all analyzed phases of decreasing Mg# 460 with increasing distance to the SZB (Figs. 10,11). Coupled to the decrease in Mg# are in most cases an increase in $TiO_2$ and a decrease in $Cr_2O_3$ (Figs. 10,11). In the following, deviations from this trend and phase specific geochemical variations are presented. Detection limits (S1), the complete microprobe data (S4) and additional graphs (S5) are attached.







**Fig. 10. EPMA data plots of ortho- and clinopyroxene porphyroclasts and of neoblasts situated at 96 m, 177 m and 290 m to the**
**SZB. Neoblasts were analyzed in cpx/opx porphyroclast tails and in the mixed matrix. A/B: Mg# against the distance to the SZB.**
**C/D: TiO₂ against Mg#. Pyroxenes of clinopyroxene-amphibole vein (Fig. 9) are indicated. E/F: Cr₂O₃ against Mg#.**

### 4.2.1 Orthopyroxene

All analyzed orthopyroxenes have with Mg#s (molar Mg/(Mg+Fe)) exceeding 0.89 enstatitic compositions (Fig. 10). In

general, neoblasts of tails and in the mixed matrix have lower $Cr_2O_3$, $Al_2O_3$ and $TiO_2$ abundances than opx porphyroclasts of

the same sample (Fig. 10). The decrease in Mg# with increasing distance to the SZB is most prominent in opx porphyroclasts

but also present for all neoblasts. The complete range of this trend is from Mg# 0.89 at 290 m distance to Mg# 0.91 at 90 m

distance to the SZB. The Mg# decrease (increase in FeO) is coupled with an increase of $TiO_2$ and a slight decrease of $Cr_2O_3$

(Fig. 10C/E).

### 4.2.2 Clinopyroxene

All analyzed clinopyroxenes have a diopsitic composition. For each analyzed sample, clinopyroxene porphyroclasts have in

general lower Mg#s and higher $Al_2O_3$ abundances than associated neoblasts. For $Na_2O$, CaO, $Cr_2O_3$ and $TiO_2$, systematic

differences between neoblasts and porphyroclasts of a given sample are not present (Fig. 10, S4). However, the neoblasts

have a bigger scatter in their composition of these oxides. For clinopyroxene, the decrease in Mg# is with a range from Mg#

0.89 (290 m) to Mg# 0.93 (90 m distance to the SZB) more pronounced than for orthopyroxene (Fig. 10). Like

orthopyroxene, the decrease in Mg# is coupled to a decrease in $Cr_2O_3$ and an increase in $TiO_2$ (Fig. 10D/F). Additionally,

$Na_2O$ increases and CaO decreases with decreasing Mg# and increasing distance to the SZB. Clinopyroxene neoblasts from a

crosscutting amphibole-pyroxenitic vein deviate significantly from all other analysis by markedly lower $Al_2O_3$ and $Na_2O$

abundances and increased CaO (Fig. 10, S5).

### 4.2.3 Olivine

All analyzed olivines have a forsteritic composition. Olivine neoblasts follow the trend of decreasing Mg# with increasing

distance to the SZB independent from the microstructural domain (Fig. 11A). However, at 290 m distance to the SZB one

group of olivine mixed matrix neoblasts tends to higher Mg#s (Fig. 11A). Yet, with lower Mg#s only present in distal

samples, the decrease of the Mg# seems to strictly depend on the distance to the SZB. CaO and NiO abundances do not vary

(Fig. 11D). Most of the $Cr_2O_3$ and all $TiO_2$ measurements lie beneath the detection limit and are therefore excluded from

further analysis (S5).

### 4.2.4 Amphibole

All amphiboles are Ti/Cr-rich pargasites with in general variable abundances of $K_2O$ (range 0-0.78 wt.%), $Cr_2O_3$ (range

0.19-1.7 wt.%) and $TiO_2$ (range 0.66-3.76 wt.%) (Fig. 11, S5). Apart from one measurement carried out on a sample situated

at 90 m distance to the SZB, all amphiboles follow the trend of decreasing Mg# with increasing distance to the SZB (Fig.





11B). Like both pyroxenes, $Cr_2O_3$ abundances decrease and $TiO_2$ abundances increase with increasing Mg# (Fig. 11E/G). For $TiO_2$, four measurements show deviations from this trend by lower abundances. There are no systematic differences between amphiboles associated to ortho-, clinopyroxene tails or the mixed matrix. $Na_2O$ and CaO abundances are except for four/three measurements constant for all samples and all microstructural domains (S5).

### 4.2.5 Spinel

Most spinels follow the trend of decreasing Mg# and increasing $TiO_2$ with increasing distance to the SZB (Fig. 11C). However, coarse grained spinels (ECD ~1 mm) associated with pyroxenes in kelyphitic intergrow at 290 m distance to the SZB show Mg#s shifted to higher values (Fig. 11C). Additionally, spinels associated to cpx neoblast tails have in each sample the highest Mg# (Fig. 11C). High Mg#s in these spinels are related to low $TiO_2$ and $Cr_2O_3$ values (Fig. 11F/H). In contrast to amphibole and both pyroxenes, $Cr_2O_3$ abundances increase with increasing distance to the SZB (Fig. 11H).

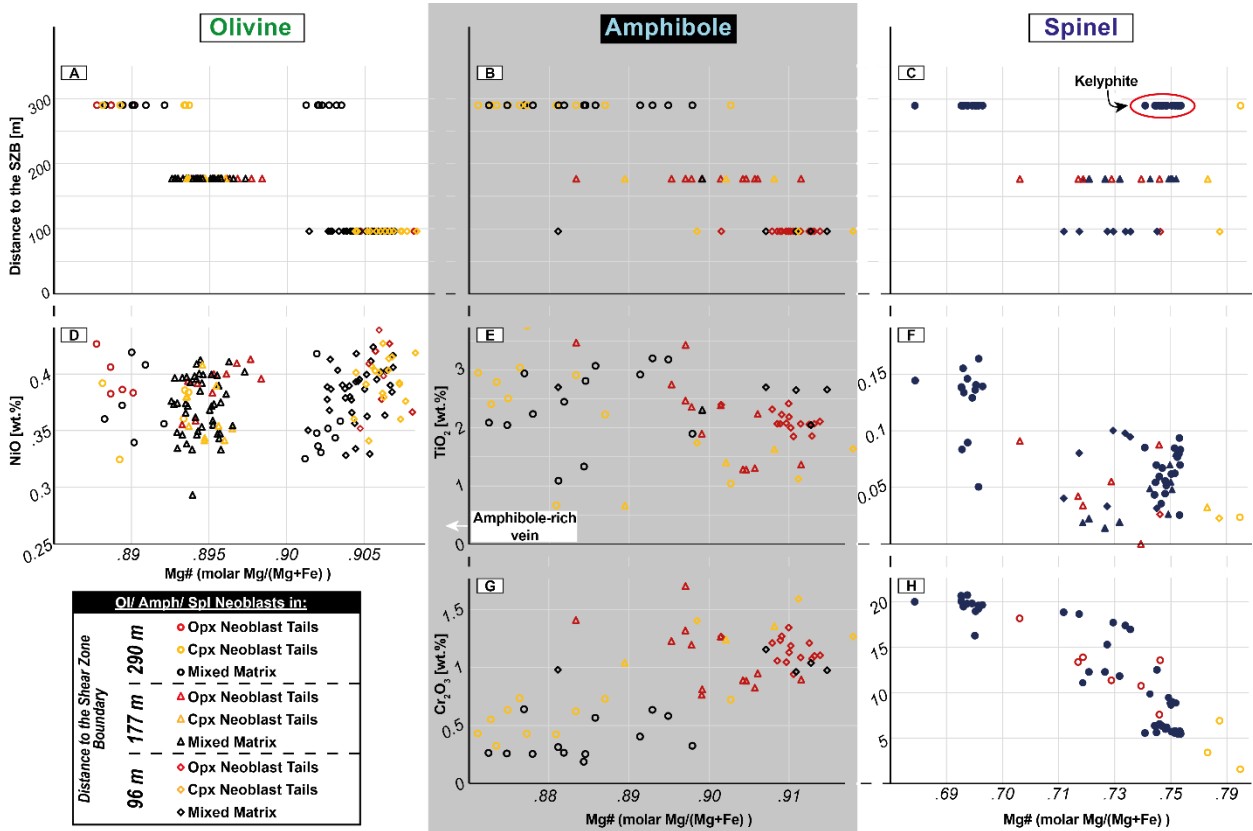

**Fig. 11. EPMA data plots of olivine, amphibole and spinel neoblasts of cpx/opx porphyroclast tails and the mixed matrix at 96 m, 177 m and 290 m distance to the SZB. A/B/C: Mg# in dependence of the distance to the SZB. D: Mg# against NiO wt.% for olivine neoblasts. E/F: Mg# against $TiO_2$ for amph and spl neoblasts. G/H: Mg# against $Cr_2O_3$ for amph and spl neoblasts.**



## 5 Discussion

### 5.1 Microstructural implications for the shear zone evolution

#### 5.1.1 Mixed matrix

As the dominant microstructure of all analyzed samples, from 29 to 700 m distance to the SZB, the mixed matrix is of particular importance for understanding the formation and evolution of the shear zone. Beside grain shape and grain size, most microstructural parameters of the mixed matrix are constant over the entire transect (next paragraph, Fig. 4). Differences and constants suggest a uniform genesis but different deformation histories of mylonites and tectonites:

Following the consistencies, several lines of microstructural evidence indicate a common formation process for the mixed matrix of tectonite and mylonites. The distribution between interstitial, secondary grains (pyroxenes, spinel) and coarser pyroxenes with highly lobate grain boundaries is alike in all mixed matrix domains. Accordingly, the presence of interstitial neoblasts with low dihedral angles and neoblasts along boundaries of coarse olivine is already present in the matrix of tectonites, which is in turn their dominant microstructure. Interstitial grains with low dihedral angles, embayments in coarse orthopyroxene and highly irregular grain boundaries of coarse pyroxenes and olivines are microstructural indications for secondary crystallization of melts and melt-rock reactions (Dijkstra et al., 2002; Stuart et al., 2018; Suhr, 1993). Following Soustelle et al. (2009), interstitial pyroxenes are therefore thought to have crystallized from a Si-rich grain boundary melt. Constant phase assemblage and constant phase abundances in the mixed matrix of mylonites and tectonites suggests that both units were affected similarly by the melt. The commonly found irregular, highly lobate boundaries between olivine and orthopyroxene suggest a reaction already evoked by Dijkstra et al. (2002) for melt assisted shearing in the Othris peridotite:

Orthopyroxene + low-Si melt ↔ Olivine + high-Si melt

(Reaction I, Dijkstra et al. (2002))

The direction of this reaction was shown to be dependent on local stress variations: Orthopyroxene decay was dominantly observed in stress shadows and thereby on boundaries at a high angle to the foliation. On the other hand, pyroxene precipitation mainly occurred along boundaries oriented subparallel to the foliation (Dijkstra et al., 2002).

Main differences between mylonitic and tectonic mixed matrix are the grain shapes and, subordinate, also the grain sizes. The tectonitic mixed matrix is characterized by small, equiaxial, interstitial grains of both pyroxenes and spinel between coarse pyroxenes and olivines. Neoblast formation in both the tectonic mixed matrix and around orthopyroxene porphyroclasts at the tectonite-mylonite transition, show weak dependence on the foliation. Additionally, tectonic mixed matrix orthopyroxene neoblasts have a CPO with [001] subperpendicular to the foliation, which is atypical for a deformation-imposed CPO. Distributed neoblast precipitation, equiaxial neoblast grain shapes and irregular CPO indicate weaker deformation in tectonites and distal mylonites. Olivine, on the other hand, with its strong A-type CPO and lobate grain boundaries was affected by dislocation creep in form of dynamic recrystallization.





For the mylonitic unit, neoblast and pyroxene porphyroclasts show similar irregular grain boundaries but a different shape and size of neoblasts. There is a clear preferred orientation of film-like pyroxenes along grain boundaries subparallel to the foliation, which was also observed in peridotite mylonites from the Othris shear zone (Dijkstra et al., 2002). Following

Dijkstra et al. (2002), these pyroxenes are interpreted as "high stress" precipitates of reaction I. Highly irregular phase boundaries between pyroxene and olivine sub-perpendicular to the foliation in places present in the mixed matrix and dominant in tails of pyroxenes indicate pyroxene melt-rock reactions as "low stress" variant of reaction I (Dijkstra et al., 2002). The strong CPO of olivine and of both pyroxenes suggest dislocation creep as the main deformation mechanism (Johanesen and Platt, 2015; Précigout and Hirth, 2014). In olivine, the dominant A type indicates slip on (010) in [100] (e.g.,

Karato et al. 2008). In orthopyroxene, slip on (100) or on (010) is dominant with both directed towards [001] (Ohuchi et al., 2011; Ross and Nielsen, 1978). Towards the SZB the dominant olivine CPO changes from an A-type CPO, indicative for low water and intermediate stress conditions, to an AG-type or occasionally a B-type CPO, indicative for increased water content and high stress (e.g., Jung, 2017). The increased presence of olivine B-type CPOs towards the SZB was formerly interpreted to result from grain boundary sliding (GBS) rather from a change in the dominant slip system (Précigout and

Hirth, 2014). However, the CPO of pyroxene neoblast tails discussed in the next chapter does not support this assumption. Deformation in the shear zone was probably enhanced by both the presence of melt during the early stages of shearing (e.g., Hirth and Kohlstedt, 1995) and the delimiting effect of secondary crystallized pyroxenes on the grain growth by pinning (e.g., Linckens et al., 2011). The resulting grain size reduction favours the activation of a grain size sensitive deformation mechanism in the shear zone, which is documented by the piezometric data of Johanesen and Platt (2015) and Précigout et

al. (2007). In contrast to GBS accommodated by dislocation creep (DisGBS, Hirth and Kohlstedt, 2003) suggested by Précigout et al. (2007) and Précigout and Hirth (2014), Johanesen and Platt (2015) favoured dislocation creep with a grain size sensitivity given by grain boundary migration as dominant recovery mechanism (DRX creep, Platt and Behr, 2011). Because microstructural evidence for both mechanisms are present (GBS: grain and phase boundary alignments; DRX creep: lobate grain boundaries) and DRX creep and DisGBS are dominant under approximately the same conditions of grain size

and shear stress (Johanesen and Platt, 2015), neither mechanism can be excluded by this study. However, the elongation of all present grains and the strong CPO for all phases demonstrates that the deformation was accommodated by dislocation creep in the entire mylonitic matrix. Even though the activity of GBS is supported by straight grain boundaries and smaller grain sizes in the mylonites, phase mixing in the mixed matrix is a consequence of crystallization of interstitial pyroxenes rather than by extensive GBS as proposed by Précigout et al. (2007).

**5.1.2 Pyroxenites and pyroxene porphyroclast neoblast tails**

Due to their microstructural and geochemical similarities both pyroxenes (opx and cpx) will be discussed together. In the tectonite, clinopyroxene porphyroclasts are often associated with pyroxenitic layers, which show a coarse-grained intergrowth of both pyroxenes, olivine and spinel. Garrido and Bodinier (1999) interpreted these websteritic layers as formed at the expense of garnet-bearing pyroxenites by melt-rock reactions. The kelyphitic structures in pyroxenite layers of





mylonites, also described by Van Der Wal and Vissers (1996), corroborate that these assemblages represent at least partially breakdown products. However, the replacement of garnet-bearing by websteritic assemblages, which is in our samples present up to the tectonite-mylonite transition, was so far associated with the melting/recrystallization front (Garrido and Bodinier, 1999). In the tectonite regime, straight grain boundaries with 90° angles within the pyroxenites suggest that these websteritic assemblages were partly annealed after having replaced garnet-bearing assemblages.

Already in the distal part of the mylonite zone, these pyroxenite layers are affected by pinch-and-swell structures, which result from boudinage. In the same samples, the formation of neoblast tails of cpx porphyroclasts and fine-grained patches of neoblasts bordering irregular, lobate boundaries of opx porphyroclasts with indentations of all neoblast phases indicate reactions. Towards the SZB the proportion of intact pyroxene porphyroclasts to reacting porphyroclasts decreases. However, elongated, mostly "retort shaped" (Johanesen and Platt, 2015) and/or stable opx porphyroclasts suggest that deformation of

opx was accommodated rather by intragranular deformation than by neoblast-formation like for most cpx porphyroclasts. Nevertheless, the phase assemblage of neoblasts tails (cpx, opx, ol, spl, amph) remains constant for tails of both pyroxenes in all mylonitic samples pointing to a common reaction as neoblast formation process for both pyroxenes. Indentations of amphibole into pyroxene porphyroclasts underline that amphibole is part of the primary neoblast assemblage. Pargasitic amphibole has been shown to be stable up to ~3.8 GPa at 1000 °C with its stability strongly depending on the amount of bulk

$H_2O$ (Mandler and Grove, 2016). Accordingly, pargasite-bearing peridotites have been shown to be stable in peridotite shear zones at similar, syn-kinematic PT-conditions to those present in Ronda (Garrido et al., 2011; Johanesen et al., 2014: 1.95-2/ 1.5 GPa, 800-900 °C; (Hidas et al., 2016; Tholen et al., 2022). The common association of pyroxenes, olivine and amphibole, indentations of amphibole into pyroxene porphyroclasts also reported by Van der Wal (1993) and the observation that spinel is less abundant in areas with amphibole and vice versa suggests a reaction of pyroxenes, spinel and amphibole.

The replacement of clinopyroxene and spinel by amphibole in peridotites is commonly referred to metasomatic reactions (e.g., Blatter and Carmichael, 1998; Bonadiman et al., 2014; Ishimaru et al., 2007). Hydrous melts were observed forming amphibole at the expense of primary orthopyroxene, olivine and clinopyroxene (Rapp et al., 1999; Sen and Dunn, 1995). In their study of xenoliths from antarctica Coltorti et al. (2004) suggested a melt-assisted reaction that crystallized amphibole at the expense of clinopyroxene and spinel shortly (few thousand years) before their uplift. Their model implies a two-stage

melt-rock evolution with an initial crystallization of pyroxenes, olivine and spinel succeeded by the secondary crystallization of amphibole. However, the composition of associated glass suggests that the metasomatizing agent was a Na-alkali silicate melt. For Ronda, constant $Na_2O$ abundances for clinopyroxene clasts and neoblasts and Ti/Fe enrichment for pyroxenes suggest a Fe-Ti-rich silicate melt. According to the experimental results of Wang et al. (2021), the composition of the crystallizing amphibole varies greatly depending on the tectonic setting, metasomatic melt and peridotite composition.

According to Coltorti et al. (2007), relatively low Mg# and high $Na_2O$ and $TiO_2$ abundances of the analyzed amphibole indicate in this regard a supra-subduction zone metasomatism.

Over the entire mylonitic area, independent on the distance to the SZB, olivine CPOs from pyroxene neoblast tails are predominantly B- or-AG type. Pyroxene tail microstructures, which include, due to the scanning arrangement, areas of or





transitions to the surrounding matrix, tend to have AG- or A-type olivine CPOs. On the opposite, a stronger B-type is
commonly bound to a well-defined neoblast tail without large amounts of the surrounding matrix highlighting the relation
between CPO-type and microstructural location. Accordingly, the girdle distributions of olivine's [100] and [001] within the
foliation plane present in the AG-type could result from a mix of A- and B-type CPOs. Amphibole, which is concentrated in
pyroxene neoblast tails and often associated with clinopyroxene, indicates higher OH abundances in these tails. This in turn
corroborates the association of B-type CPO to increased concentrations of H/Si (Jung et al., 2006; Jung and Karato, 2001;
Mizukami et al., 2004). The correlation of a stronger B-type with increased clinopyroxene abundances observed by
Précigout and Hirth (2014), which was at odds with the B-type solely dependent on the increase of GBS towards the SZB
therefore fits with both presented observations: Pronounced presence of amphibole and olivine B-type CPOs in pyroxene
neoblast tails and the preferred association of amphibole with clinopyroxene. Accordingly, the decrease of porphyroclasts
and the increase in pyroxene neoblast tails towards the SZB leads to an increase of olivine neoblasts with B-type orientation.
However, the formation of olivine B-type CPOs by GBS in the mixed matrix close to the SZB (< 100 m) suggested by
Précigout and Hirth (2014) cannot be ruled out. Although multiphase mixtures crystallized in the metasomatic neoblast tails
of pyroxenes, no strain localization as reported for pyroxene reaction tails in other peridotite shear zones occurred in these
microstructural domains (Hidas et al., 2013b; Tholen et al., 2022). The main reason for the lack of strain localization might
be that all microstructural domains have similar amounts of phase boundaries and similar grain sizes. Therefore, no strain
partitioning between the mixed matrix and the tails associated with a switch to a grain size sensitive deformation mechanism
was achieved (e.g., Rutter and Brodie, 1988).

### 5.1.3 Grain size evolution

The change in the overall microstructure in the shear zone, interpreted as a continuous decrease in grain size towards the
SZB by several authors (e.g., Obata, 1980; Précigout et al., 2007; Van Der Wal and Vissers, 1996) should be strongly related
to the change in grain shape and grain size of the mixed matrix between tectonites and mylonites described above. In
agreement with Johanesen and Platt (2015), who reported a rather constant grain size of recrystallized olivine (~130 µm)
with regional variations in mylonites and tectonites, neoblast grain sizes of all phases and from all microstructural domains
stay constant over the entire mylonitic shear zone. Accordingly, and following Johanesen and Platt (2015), the trend of
decreasing total grain size with decreasing distance to the SZB reported by previous studies is explained by the increasing
amount of neoblasts rather than by a systematically change in their grain size. Only the outermost, tectonic sample shows
larger grain sizes in the mixed matrix domain, whereas grain sizes in pyroxene neoblast tails remain constant but have a
bigger spread. Furthermore, an excursion towards bigger grain sizes (deviation ~50 µm) comparable to the variations found
by Johanesen and Platt (2015) is present in the mixed matrix and in orthopyroxene neoblast tails at ~290 m distance to the
SZB. Due to a smaller number of analyzed microstructures in these samples, the excursion is only rudimentary present for
neoblasts of cpx tails. As these local variations are present for most phases in multiple microstructural domains, they indicate
variations in differential stress independent of the distance to the SZB. Corroborating the results of Johanesen and Platt





(2015), no gradual increase of stress towards the SZB is indicated in our data, which would result in a gradual decrease in grain size. In contrast to the average recrystallized olivine grain size of ~130 µm from Johanesen and Platt (2015) our average grain sizes are smaller for all phases and all microstructural domains (< 100 µm). Reporting only the grain size of

recrystallized olivine, which was excluded for matrix microstructures in this study due to uncertainties caused by serpentinization, our grain sizes are not consistent to those of Johanesen and Platt (2015), which could explain this discrepancy. However, the increase in grain size in the tectonite unit and its coarser, less deformed microstructures corroborate different deformation histories for mylonites and tectonites as described above (section 5.1.1). Differences in the shape of primarily the neoblasts of the mixed matrix are therefore thought to be dependent on the strain of the specific unit.

To that effect, the elongation of neoblasts in the mylonitic mixed matrix, the elongation of opx porphyroclasts and the elongation of pyroxene tails stretched in the foliation are interpreted as increased strain towards the SZB at almost constant stress (= constant grain sizes). The increase in strain could be either due to an increase of the strain-rate (Johanesen and Platt, 2015) or a longer-lasting deformation in mylonites, which will be addressed below.

## 5.2 Melts in the shear zone

Increasing evidence for metasomatic processes and melt-rock interactions were reported closer and closer to the NW boundary of the Ronda peridotite during the last decades. The so called "recrystallization front", which separates the spinel tectonites from the coarse grained peridotites, was revealed as melting front (Lenoir et al., 2001; Soustelle et al., 2009; Van Der Wal and Jean-Louis, 1996). Soustelle et al. (2009) confirmed that Si-rich melts fertilized the spinel tectonites of the NW Ronda shear zone up to 1.5 km ahead of this melting front. According to these authors, early melt pulses lead to pyroxene

and spinel crystallization as irregularly shaped grains, whereas late stage, second-order percolation of evolved melt caused the crystallization of interstitial, undeformed pyroxenes and spinel with a strongly enriched LREE chemistry. The melt is postulated to be derived by partial melting (2.5-6.5 % extraction) of the coarse grained peridotites (Lenoir et al., 2001). Beside melt-assisted crystallization, partial melting of garnet pyroxenite layers in the tectonites and their increasing conversion into spinel pyroxenites towards the melting front can also be placed in the context of melt percolation and the

steep thermal gradient from the melting front to the SZB (Garrido and Bodinier, 1999). However, even though the deformation in the entire shear zone (tectonites + mylonites) shows continuous orientations of foliation and lineation and the timing of the melt percolation is postulated to be syn- to late-kinematic (Soustelle et al., 2009), the westernmost mylonite unit was so far considered either to be completely melt-free (Précigout et al., 2007; Soustelle et al., 2009) or melt-absent during the deformation (Johanesen and Platt, 2015). Based on our combined microstructural and geochemical investigations

we can extend the sphere of melt influence into the mylonites and up to the NW edge of the Ronda peridotite.





**Fig. 12. Mg# data of spinel tectonites (Soustelle et al., 2009) and of spl/grt mylonites (this study) vs. distance to the SZB. Arrows indicate trend of re-fertilization. Hatched area: geochemical signature of melt in tectonites; Crosshatched area: geochemical signature of melt in mylonites. Mg# of opx (A), cpx (B), olivine (C) and spl (D) plotted against the distance to the SZB. Location of**
**the studied area by Soustelle et al. (2009) is indicated in Fig. 1.**

Following Soustelle et al. (2009), we interpret the interstitial pyroxenes and spinel of the mixed matrix in tectonites and mylonites as precipitates from percolating, re-fertilizing melt. The associated melt-rock reactions are geochemically mainly characterized by an increase in FeO (= decrease in Mg#) and $TiO_2$ for olivine, pyroxenes, spinel and amphibole towards the melting front, with increasing distance to the SZB respectively. The continuous geochemical trends and the presence of
interstitial spinel and pyroxene neoblast between olivine crystals in the mixed matrix of both, mylonitic and tectonic samples, suggest that the re-fertilizing melts leading to pyroxene crystallization were present in the entire transect. In contrast to Soustelle et al. (2009) who reported a fertilization only 1.5 km ahead the melting front and therefore leaving the mylonitic shear zone unaffected, our data traces the re-fertilizing melts up to the SZB. In Fig. 12, data from Soustelle et al. (2009) was put in correlation to the distance to the SZB and added to the data presented here. Indicated by the grey arrows is
the geochemical trend of increasing Mg# with decreasing distance to the SZB. It is obvious that the data of Soustelle et al. (2009) and our data follow this trend (Fig. 12). Anyhow, an offset is present between both data sets with data of Soustelle et al. (2009) staring to decrease from higher Mg# (~0.92) than the analyzed data herein (~0.89) (Fig. 12). This offset in Mg# between the, in regard to the SZB, distal tectonites (Soustelle et al., 2009) on the one hand, and the mylonites and tectonites closer to the SZB on the other hand (analyzed herein) is potentially the geochemical im- and overprint of multiple re-
fertilization events (Fig. 12). The re-fertilization documented in our samples and characterized by lower Mg#s is in this regard assigned to an early melt impulse affecting the entire area between melting front and SZB, present day tectonites and mylonites respectively. As samples remote from the SZB (>1150-1200 m) do not follow the geochemical trend present in those analyzed herein but rather follow a new, second-order trend beginning with increased Mg# for all analyzed phases (Fig. 12) a late-stage melt infiltration overprinting this area close to the melting front is most plausible. Interestingly,
samples with a distance of ~1150-1200 m to the SZB from Soustelle et al. (2009) record with their wide range of Mg#s apparently both re-fertilization events. Further, both re-fertilizations follow the same geochemical trend and trajectory, indicating the area beneath the structurally deeper melting front as the melt source (Fig. 12). A comprehensive sketch of the interplay of microstructural evolution and melt percolation is shown in Fig. 13. The possibility of varying source rocks being responsible for the various trends is unlikely because different lithologies (lherzolites and harzburgites) and diverse
microstructures (neoblast/porphyroclasts of different domains) analyzed by Soustelle et al. (2009) and herein follow the same trend. The tendency of mixed matrix pyroxene neoblasts to lower $TiO_2$ and $Cr_2O_3$ abundances could indicate a stronger effect of diffusion on the smaller grains (Cherniak and Liang, 2012). This process could additionally be enhanced by ongoing deformation as elongated grain shape and size of the mylonitic mixed matrix suggest. The melt-infiltration in the mylonites is therefore thought to be pre- to early syn-kinematic.
As no geochemical difference is present between the various microstructures, pyroxene tail assemblages are thought to be affected by the re-fertilization event like the mixed matrix. The synkinematic formation of the neoblast tails is therefore



attributed to the same deformation stage, which was initiated by the re-fertilization of the entire shear zone. Melt infiltration has been shown to significantly reduce the mechanical strength of the upper mantle (e.g., Tommasi et al., 2017). The strength contrast between melt-affected (tectonites+mylonites) and melt-free areas is a plausible reason for the localization

of deformation at the boundary between both areas. The flattening of the re-fertilization imprint (Mg#, $TiO_2$) towards the present day SZB might indicate, that this boundary coincides approximately with today's SZB. With progressive deformation, the area accommodating the deformation expands further into the melt-affected region, towards the melting front respectively, forming the mylonitic unit. As the strain localization starts at the boundary of the melt-affected area, here the deformation is active for the longest time resulting in maximum degree of finite strain. Hence, porphyroclast elongation

increases towards the boundary (present day SZB). Additionally, the increase in pyroxene porphyroclast neoblast tails with concurrent decrease of porphyroclasts towards the SZB might indicate a strain dependence. De Ronde and Stünitz (2007) reported a positive feedback between deformation and reactions in their experiments for the transition from plagioclase to spinel in olivine+plagioclase aggregates. An enhanced nucleation reaction rate was here explained by increasing deformation-induced defects in the reactant and the deformation-induced transportation of neoblasts away from the reaction

interface, which thereby maintains a high chemical potential. For Ronda, a similar mechanism could clear the melt-porphyroclast interface of neoblasts and thereby form the, in the foliation elongated, neoblast tails. With constant stresses and constant dominant deformation mechanism(s) operating in the melt-affected area neoblast grain sizes are kept nearly constant.







**Fig. 13.** Sketch of microstructural, shear zone and Mg# evolution of the NW Ronda shear zone. (1): Layered lithospheric mantle: Equilibrated assemblage of coarse olivines and pyroxenes cut by grt-bearing pyroxenite layers; homogenous Mg#. (2) and (3): Melt infiltration of the entire shear zone. Formation of mixed matrix by crystallization of interstitial pyroxenes in the entire shear zone (2). Areas little effected by melt form Ol-rich matrix. Apart from distal pyroxenites, grt is replaced by kelyphitic opx+cpx+spl. "Melting front" as origin of the re-fertilizing melts is shown as hatched area in the shear zone evolution. Fe- and Ti-rich melt shifts Mg# of pyroxenes and olivine towards lower values. Increased deformation leads to elongation of olivine and wedge-like pyroxenes in the mylonites (3). Pyroxene porphyroclasts form amph-bearing neoblast tails. Ol-rich matrix form lenses in the mixed matrix. (4) Melt infiltration in tectonites documented by Soustelle et al. (2009) with geochemical and textural overprint. Coarse, less-deformed grains with equiaxial interstitial pyroxenes. Mg# shifted towards higher values.







### 5.3 Late-stage fluid infiltration

For the sake of completeness, the fluid-infiltration, documented in several samples, will be addressed in the following section (see also Fig. 9). The crosscutting of amphibole-filled cracks of entire cpx porphyroclasts, the replacing of cpx exsolution lamellae in opx porphyroclasts by amphibole described by Obata (1980) and the formation of amphibole and clinopyroxene rich veins oblique to the formation indicate a late-stage fluid infiltration without relation to the melt infiltration and deformation processes discussed above. Since these observations were primarily made in mylonites close to

the SZB, a fluid infiltration originating from the adjacent metasedimentary Jubrique unit seems plausible. Lower Ti abundances for amphibole, clino- and orthopyroxene neoblasts and amphibole Mg#s not comparable (<0.86) to those of other microstructural domains corroborate an independent formation process. Interestingly, the formation of serpentine seems to follow these structures when present.

### 5.4 Reactions and deformation

Like for most studied upper mantle shear zones, the results presented for the Ronda shear zone point to a key-role of reactions in the evolution of upper mantle shear zones (e.g., Dijkstra et al., 2004). A comparison between these studies suggests that the impact of reactions on the evolution of shear zones depends rather on the timing than on the type of reaction:

Tommasi et al. (2017) have shown that hydrous Si-rich melts significantly affect the rheological strength of the upper mantle

and favour a strain localization in the melt-effected region. Additionally, melt-rock reactions in low strain microstructures of the Lanzo shear zone indicate melt-presence during initial shearing (Kaczmarek and Müntener, 2008). Beside phase mixing by crystallization of pyroxene neoblasts interstitially and at the reacting boundaries of coarser olivine in combination with the activity of a grain size sensitive creep mechanism (Hirth and Kohlstedt, 2003; Platt and Behr, 2011) an additional effect is the reduction of the viscosity by "wetting" of the grain boundaries (e.g., Hirth and Kohlstedt, 1995). As these effects are

solely dependent on the presence of melt, they also are most likely decisive for early localization of strain in the upper mantle.

For syn-tectonic, high stress conditions during the later stages of the shear zone evolution, metasomatic and metamorphic reactions were shown to be decisive for the formation of ultramylonitic neoblast assemblages either in pyroxene porphyroclasts tails or in ultramylonitic bands: In the shear zones of Othris and Erro Tobbio melt-rock reactions formed

ultramylonitic, mixed tails dominated by pyroxene and olivine (Dijkstra et al., 2002; Linckens and Tholen, 2021). Metamorphic reactions in relation to the phase transitions of garnet, spinel and plagioclase triggered reactions at pyroxene porphyroclasts and the formation of ultramylonitic assemblages in shear zones of the Uenzaru peridotite complex, the Turon de Técouère peridotite body and the Lanzo peridotite massif (Furusho and Kanagawa, 1999; Newman et al., 1999; Tholen et al., 2022). Fluid presence enhancing dissolution-precipitation creep and leading to the formation of ultramylonites was

reported for shear zones at the transition from plagioclase to granular peridotite in central Ronda (Hidas et al., 2016) and in





the Anita Peridotite (Czertowicz et al., 2016). Phase mixing with amphibole and/or chlorite in ultramylonitic assemblages was reported for Erro-Tobbio (Hoogerduijn Strating et al., 1993; Linckens and Tholen, 2021) and the Shaka and Prince Edward transform fault (Kohli and Warren, 2020; Prigent et al., 2020). Diffusion creep and GBS as dominant deformation process in these ultramylonitic assemblages weaken the rheology significantly leading to further strain localization in the shear zones if the ultramylonitic areas are interconnected (e.g., de Ronde et al., 2005).


To summarize: Reactions weaken the upper mantle and lead to strain localization. The degree of strain localization seems to dependent on the timing of the reaction in the course of the shear zone evolution, not on the nature of the reaction itself. In Ronda, extensive melt infiltration localized the deformation over a km-scale area in tectonites and mylonites and thereby shaped the shear zone. High mixing intensities and resulting homogenous grain sizes in the mylonitic mixed matrix ensured that no further strain localization did occur in porphyroclast's reaction tails. In the shear zones of Othris (Dijkstra et al., 2002), Erro-Tobbio (Linckens and Tholen, 2021), Uenzaru (Furusho and Kanagawa, 1999), Turon de Técouère (Newman et al., 1999) and central Ronda (Hidas et al., 2016) synkinematic melt/fluid-assisted and/or metamorphic reactions under high stress conditions led to the formation of mixed ultramylonitic bands. In these bands strain is further localized in the dm- to cm-scale by a switch to diffusion creep as dominant deformation mechanism.


**6 Conclusion**


Microstructural and geochemical analysis of the major microstructural domains of the NW Ronda shear zone document a multistage melt derived re fertilization with a fundamental impact on strain localization and the formation of the km-scale shear zone. The interaction of melt and deformation in the course of the shear zone evolution can be illustrated by the following bullet points (see also Fig. 13):

1) The initial, largely undeformed rock belongs to a coarse-grained lithospheric mantle (ol+opx+cpx) with a relatively homogenous composition and layers of grt-bearing pyroxenites.


2) Melt-infiltration re-fertilizes the complete present-day shear zone. Strain localization is initiated by melt-enhanced deformation and grain size reduction by the crystallization of interstitial pyroxenes and the subsequent activation of a grain-size-sensitive deformation mechanism. Potential dominant deformation mechanisms are DRX creep and/or DisGBS. The change in composition (Mg#↓, Fe↑, Cr↓) of pyroxenes and olivine and its fading towards the shear zone boundary suggest that the area affected by strain localization is strongly dependent on the presence of melt. At the shear zone boundary, areas little effected by melt preserve garnets (Garrido et al., 2011).


3) Increased deformation localized in the melt-affected areas led to the formation of mylonites. With constant stress over the entire melt-affected area but an earlier onset of deformation at the shear zone boundary a strain gradient is formed. Its microstructural evidence are the increasing elongation of orthopyroxene porphyroclasts, neoblasts and the boudinage of pyroxenitic layers towards the shear zone boundary. Furthermore, deformation-induced reactions led to an increase of




neoblast formation in reaction tails and the decrease of pyroxene porphyroclasts in mylonites situated close to the shear zone boundary.

4) Secondary melts infiltrate the tectonites up to ~1.5 km ahead of the melting front (Soustelle et al., 2009). Microstructures are affected by recrystallization and increase in Mg#. The geochemical signature formed in step 2 is overprinted. Equiaxial pyroxenes with low internal deformation corroborate a late-kinematic timing (Soustelle et al., 2009).

**Supplementary data (attached as .zip file)**

S1 - EPMA measurement settings and detection limits.

S2 – Microstructural data of all analysed (EBSD) microstructures.

S3 – CPO overview of all analysed (EBSD) microstructures.

S4 – Complete EPMA data.

S5 – EPMA additional graphs. Additional graphs for clinopyroxene, olivine and amphibole.

**Author contribution**

**Sören Tholen:** Conceptualization, Methodology, Software, Validation, Formal analysis, Investigation, Data Curation, Writing, Visualization, Project administration. **Jolien Linckens:** Conceptualization, Validation, Methodology, Resources, Project administration, Supervision, Funding acquisition. **Gernold Zulauf:** Resources, Writing (Review), Funding acquisition.

**Competing Interests**

The authors declare that they have no conflict of interest.

**Acknowledgments**

For fruitful discussions and comments along the way, we want to thank Alan Woodland, Catharina Heckel, Reiner Kleinschrodt and Marina Kemperle. Illuminating field information were provided by Jacques Précigout and Carlos J. Garrido. Without the help of Thomas González and his team at the Sabinillas Bookstore, most of our samples would have fallen victim to dodgy transport companies and COV-19 restrictions, thank you very much! For superb sample preparation Maria Bladt and Nils Prawitz are to be thanked. We are grateful for the collaboration with the Institute of Geology and Mineralogy Cologne and want to thank again Reiner Kleinschrodt, Patrick Grunert and Hannah Cieszynski. This project was made possible by funds of the Deutsche Forschungsgemeinschaft (DFG) [LI 2888/2-1].



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
