# Peer review of "Melt-enhanced strain localization and phase mixing in a large-scale mantle shear zone (Ronda peridotite, Spain)"

_EGUsphere, 2022_

## Author Comment (AC2)

Dear Editors, dear Andréa Tommasi, dear Jacques Précigout,

First of all, the authors want to thank both reviewers, Andréa Tommasi and Jacques Précigout, for their detailed and constructive comments, which help to improve the manuscript. The authors decided to reply in one statement because the major remarks of both referees largely overlap or complement each other. In general, all proposed changes and comments of both reviewers were considered in the revised version of the manuscript. The changes tracked by line number are in the second part of this reply. However, at first we will briefly comment on the main points of criticism.

According to both reviewers, the three main points of criticism are:

(1) The nature of metasomatism: Refertilizing or late stage, fluid-rich melt?
(2) The timing of the metasomatism and its effect on deformation.
(3) The insufficient quantification of the olivine grain size.

The nature of metasomatism:

We agree with both reviewers, that the irregular grain/phase boundaries and grain shapes as well as extensive phase mixing are robust microstructural evidence for metasomatism in the entire NW Ronda shear zone. A metasomatism of parts of the investigated mylonite unit by refertilizing melts was postulated by Soustelle et al. (2009). This process was adapted in the original manuscript for our transect. However, both reviewers reject the interpretation of a refertilizing melt because of low syn-kinematic temperature estimates (800–900 °C, 1.95–2.00 GPa (Garrido et al. (2011)) and the missing modal change to increased fertile components. The authors agree with both objections and discuss the nature of the metasomatism in the reviewed version by taking into account the suggestions made by the reviewers. Following the annotations made by Andréa Tommasi, the microstructural similarity as well as the matching PT-estimations for the grt/spl-mylonites from Rondas counterpart from the Morrocan limb of the Gibraltar arc, the Beni Bousera peridotite massif, point to a consistent genesis. Frets et al. (2012, 2014) suggested a metasomatism of small fractions of fluids or highly evolved melts, which did not reset the quilibrium temperatures in Beni Bousera. Matching all observations made in our samples, the authors agree, that the metasomatism is most likely attributable to highly evolved melt. A fluid-driven metasmomatism as proposed for the plagioclase-tectonite unit in Ronda by Hidas et al. (2016) is in the authors opinion less likely because of the low abundance of amphibole in the dominant mixed matrix and the absence of ultramylonites, which were obsereved to form by fluid-rock reactions. Based on these observations, the rewritten section 5.1 "Microstructural implications – Formation" now includes a discussion and evaluation of the different potential metasomatic agents. In this regard, the authors agree with Jacques Précigouts remark of the small geochemical data base for a geochemically based model of the shear zone's evolution. To resolve the geochemical trend in detail, an additional study would be needed with the focus on the transitional area between the mylonite and tectonite unit. According to Jacques Précigouts suggestions, the reviewed discussion was focused on the microstructural evidence.

The timing of the metasomatism and its effect on deformation:

The rewritten section 5.2 "Microstructural implications – Deformation" now discusses the timing of the metasomatic event and its effect on the deformation. Microstructural similarities especially of the film/wedge-shaped orthopyroxenes in the mylonitic part of the shear zone to mylonites and ultramylonites investigated by Dijkstra et al. (2002) and Hidas et al. (2016) indicate a syn-kinematic metasomatism with dissolution-precipitation reactions being active. This assumption is supported by Frets et al. (2014), who argued for the corresponding grt/spl-mylonites of Beni Bousera for syn- to late kinematic metasomatism. As both reviewers criticize an overinterpretation of the data in terms of the importance of the metasomatic event for the genesis of the NW Ronda shear zone, the

discussion was fundamentally shortened in this regard. Therefore, the main focus of section 5.2 lies now on the active deformation mechanisms (dislocation creep, dissolution-precipitation creep), the dominant deformation mechanism (dislocation creep) and the potential impact of phase mixing and melt presence on the deformation. The authors agree that an irrevocable argument for the trigger of the shear zone by metasomatic processes cannot be given. However, the comparison with other upper mantle shear zones (section 5.4) indicates a general strong relation between reactions and localized deformation in the upper mantle. With the data presented, the NW Ronda shear zone lines up or at least does not contradict this picture.

The insufficient quantification of the olivine grain size:

The complete data was reprocessed to quantify the original olivine grain size using the method suggested by Andréa Tommasi. The new data were added to the microstructural data of figure 3 and of supplementary data S2. However, even with a larger spread and coarser grain sizes, olivine follows the general trend of constant grain sizes in the entire mylonite unit formerly reported by Johanesen & Platt (2015). Moreover, Frets et al. (2014) report for the Grt/Spl-mylonite unit of Beni Bousera a similar range of mean olivine grain size (90-160 µm). The statistics of 7375 olivine grains analyzed in the mixed matrix and the consistency with the published data indicate a robust data set of constant olivine grain size over the entire mylonite unit with local variations but no obvious trend.
* * *
Detailed list of corrections for remarks of Jacques Précigout sorted by line numbers. In the authors answers the first line number refers to reviewed manuscript without changes marked, second line number to the version with changes marked.

Jacques Précigout:

Additional comments:

1. Better synthesizing the micro-structural features, documentations of which are a bit unbalanced with respect to chemical features.

   The complete discussion was rewritten with the focus on the microstructural implications. The authors agree, that for solid geochemical interpretation additional measurements are necessary on samples that are not present at the moment. The focus of the geochemical investigation should be on the transition between mylonites and tectonites.

2. Strengthening the point of Mg# gradient by performing chemical analyses on more samples (this would help to better document the melt-rock interactions front).

   The authors decided to focus with this manuscript on the microstructural analysis of the mylonite unit. However, agreeing with the remark a future study in the geochemistry is planned.

3. Reconsidering the main axis/interpretation of the paper by focusing on the evidence of melt-rock interactions front across the shear zone, and not speculating - although it could be discussed - on the role of melt in triggering strain localization in Ronda (based on the data presented here, it is not plausible)

   In accordance with the answer on the main points of criticism and the comments above the discussion was rewritten with the focus on the microstructural implications for the formation of and deformation in the mylonitic unit.

Minor comments to the authors

Geological setting: You could be interested in having a look at the two papers of Bessière et al. (2021) that expand on the geodynamic of the Ronda peridotite.

> Thanks for the interesting suggestion.

Line 142: underlying not underlaying.

> Changed (l. 143/ 160)

Line 159: Citing a paper rather than the PhD thesis of Dirk Van der Wal would be more appropriate here. And I think some other hypothesis (and references) need to be mentioned, including the one described in our paper (Précigout et al., 2013).

> The authors decided to delete the emplacement hypothesis from the introduction as the focus lays on the microstructures and not on the overall tectonics (l. 160/ 183). As the reviewer annotates correctly otherwise more hypotheses should be discussed.
>
> The reference was changed to the most relevant paper.

Line 167: were, not was.

> Changed (l. 168/ 192)

Line 202: 100 grains is not enough to calculate a J or Mindex (Skemer et al., 2005). You could also see our recent paper dealing with this feature (Précigout et al., 2022, sci. rep.)

> A minimum of 150 grains was set for the M- and J-Index (l. 202/ 226).

Line 204: We commonly use 10° as halfwidth angle. Otherwise, it smoothes a lot the data.

> Thanks for the remark. Using 15° as halfwidth was adapted from previous studies (Tholen et al 2022, Linckens et al 2021). For the next dataset we will compare 10° and 15°.
>
> For this dataset, the CPOs are mostly strong if present and therefore the changes in the dominant olivine CPO for example are easily recognizable.

Line 223: Plane, not plain.

> Changed, thanks (l. 229/ 257)

Line 242: This feature has been already described in Précigout et al. (2013), so it should be cited here.

> Citation was included (l. 244/ 276).

Figure 4a: Where are the olivine dots? The olivine-rich matrix represents the major part of the peridotite, so you cannot exclude it from the grain size dataset, whatever the reason.

> Now figure 3A. As both reviewers requested to include the olivine grain size, we recalculated all EBSD data to reconstruct the original grain size. The reconstruction was performed using the proposed method in MTEX (Matlab) by Andréa Tommasi.

Line 334: B-type fabric has been documented in Précigout et al. (2014), so it has to be mentioned here.

> Citation was added (l. 354/ 403).

Figure 6: The number of grains (or datapoint) has to be shown by each pole figure.

> Numbers of grains and color bars for ODFs were added to all orientation figures.

Line 387: what do you mean by « tend to be higher »? The AR is higher or not.

Changed (l. 411/ 460).

Figure 8A: Avoid writing labels up side down.

Now figure 9. Label was flipped.

Line 527: Discussing about processes of mantle refertilization, the paper of Le Roux et al. (2007) should be discussed, at least cited, somewhere.

Refertilization was discussed in section 5.1.1. The citation was added.

Line 539: what do you base on to say that this CPO is atypical ? Is there any reference that mention that.

Reference was added (l. 690/ 747). The dominant CPOs for opx in deformed mantle rocks commonly have [001] in the lineation.

Line 581: when you discuss a feature that is not described in your paper, you have to cite the publication that describe it. For instance, boudinaging of pyroxenite layers has been described in Précigout et al., 2013.

Very correct. Thanks for the annotation, citation was added (l. 628/ 684).

Line 652: Saying constant grain size is not correct here, but constant dynamically recrystallized grain size may be correct.

Section was rewritten (now 5.2)

Figure 12 and 13: to be frank, your model is very difficult to understand based on the figures. Could you please make them more clear?

Complete discussion was rewritten with the focus on the microstructural implications. The model was dismissed.

Line 701: what trend?

See comment above.

---

## Author Comment (AC3)

Dear Editors, dear Andréa Tommasi, dear Jacques Précigout,

First of all, the authors want to thank both reviewers, Andréa Tommasi and Jacques Précigout, for their detailed and constructive comments, which help to improve the manuscript. The authors decided to reply in one statement because the major remarks of both referees largely overlap or complement each other. In general, all proposed changes and comments of both reviewers were considered in the revised version of the manuscript. The changes tracked by line number are in the second part of this reply. However, at first we will briefly comment on the main points of criticism.

According to both reviewers, the three main points of criticism are:

(1) The nature of metasomatism: Refertilizing or late stage, fluid-rich melt?
(2) The timing of the metasomatism and its effect on deformation.
(3) The insufficient quantification of the olivine grain size.

The nature of metasomatism:

We agree with both reviewers, that the irregular grain/phase boundaries and grain shapes as well as extensive phase mixing are robust microstructural evidence for metasomatism in the entire NW Ronda shear zone. A metasomatism of parts of the investigated mylonite unit by refertilizing melts was postulated by Soustelle et al. (2009). This process was adapted in the original manuscript for our transect. However, both reviewers reject the interpretation of a refertilizing melt because of low syn-kinematic temperature estimates (800–900 °C, 1.95–2.00 GPa (Garrido et al. (2011)) and the missing modal change to increased fertile components. The authors agree with both objections and discuss the nature of the metasomatism in the reviewed version by taking into account the suggestions made by Andréa Tommasi. Following the annotations made by Andréa Tommasi, the microstructural similarity as well as the matching PT-estimations for the grt/spl-mylonites from Rondas counterpart from the Morrocan limb of the Gibraltar arc, the Beni Bousera peridotite massif, point to a consistent genesis. Frets et al. (2012, 2014) suggested a metasomatism of small fractions of fluids or highly evolved melts, which did not reset the quilibrium temperatures in Beni Bousera. Matching all observations made in our samples, the authors agree, that the metasomatism is most likely attributable to highly evolved melt. A fluid-driven metasmomatism as proposed for the plagioclase-tectonite unit in Ronda by Hidas et al. (2016) is in the authors opinion less likely because of the low abundance of amphibole in the dominant mixed matrix and the absence of ultramylonites, which were obsereved to form by fluid-rock reactions. Based on these observations, the rewritten section 5.1 "Microstructural implications – Formation" now includes a discussion and evaluation of the different potential metasomatic agents. In this regard, the authors agree with Jacques Précigouts remark of the small geochemical data base for a geochemically based model of the shear zone's evolution. To resolve the geochemical trend in detail, an additional study would be needed with the focus on the transitional area between the mylonite and tectonite unit. According to Jacques Précigouts suggestions, the reviewed discussion was focused on the microstructural evidence.

The timing of the metasomatism and its effect on deformation:

The rewritten section 5.2 "Microstructural implications – Deformation" now discusses the timing of the metasomatic event and its effect on the deformation. Microstructural similarities especially of the film/wedge-shaped orthopyroxenes in the mylonitic part of the shear zone to mylonites and ultramylonites investigated by Dijkstra et al. (2002) and Hidas et al. (2016) indicate a syn-kinematic metasomatism with dissolution-precipitation reactions being active. This assumption is supported by Frets et al. (2014), who argued for the corresponding grt/spl-mylonites of Beni Bousera for syn- to late kinematic metasomatism. As both reviewers criticize an overinterpretation of the data in terms of the importance of the metasomatic event for the genesis of the NW Ronda shear zone, the discussion was

fundamentally shortened in this regard. Therefore, the main focus of section 5.2 lies now on the active deformation mechanisms (dislocation creep, dissolution-precipitation creep), the dominant deformation mechanism (dislocation creep) and the potential impact of phase mixing and melt presence on the deformation. The authors agree that an irrevocable argument for the trigger of the shear zone by metasomatic processes cannot be given. However, the comparison with other upper mantle shear zones (section 5.4) indicates a general strong relation between reactions and localized deformation in the upper mantle. With the data presented, the NW Ronda shear zone lines up or at least does not contradict this picture.

The insufficient quantification of the olivine grain size:

The complete data was reprocessed to quantify the original olivine grain size using the method suggested by Andréa Tommasi. The new data were added to the microstructural data of figure 3 and of supplementary data S2. However, even with a larger spread and coarser grain sizes, olivine follows the general trend of constant grain sizes in the entire mylonite unit formerly reported by Johanesen & Platt (2015). Moreover, Frets et al. (2014) report for the Grt/Spl-mylonite unit of Beni Bousera a similar range of mean olivine grain size (90-160 µm). The statistics of 7375 olivine grains analyzed in the mixed matrix and the consistency with the published data indicate a robust data set of constant olivine grain size over the entire mylonite unit with local variations but no obvious trend.
* * *
Detailed list of corrections for comments by Andréa Tommasi, sorted by line numbers. In the authors answers the first line number refers to reviewed manuscript without changes marked, second line number to the version with changes marked.

Additional comments:

1. The introduction and discussion sections have repetitions and may be significantly shortened, so that there will be more space to present the data, which in the present form of the ms. is largely presented as Supplementary material. The section of the amphibole-bearing veins is also not essential to the article.

   Both sections were revised and reworked. Parts were shortened and the complete CPO data was included in the results section. The authors decided to leave the section on amphibole-bearing veins (4.1.3) in the manuscript to make readers aware of this feature which is potentially interesting to investigate late-stage fluid-peridotite interaction and was not described so far.

2. Please add a map of the Sierra Bermeja massif with foliations and lineations. Even if you focus on the microstructures, the structural context is important. In general, the description of the structural data is too vague. The orientation of the mylonitic foliations is much more varied then stated in l. 115 - their trend follows on average that of the limits between the tectonometamorphic domains, see maps from Darot (1973) and data reported in later studies (Obata, Van der Wal et al 1993, 1996, Soustelle et al. 2009…). Same for the lineations.

   A structural map of the Sierra Bermeja massif including foliation, lineations and major faults was added to figure 1. The studied area of Soustelle et al. 2009 was indicated in this map. The section on the structural data was rewritten to clarify the variations in the foliation/lineation and the dominant orientation of both in the area of investigation (ll. 139/ 156, ll. 225/ 251).

3. The description of the sampling referring to the shear zone boundary is not always clear. Better state that the samples were collected at increasing distance from the northeastern limit of the massif. This limit is not necessarily the limit of the shear zone as the contact between the peridotites and the Jubrique unit may have been reworked. Similarly, in line 371, the use of distal may lead to confusion.

> Shear zone boundary (SFZ) was changed to NW boundary of the Ronda peridotite massif (NW-B). The abbreviation was necessary for graph axes titles etc.. The use of "proximal" and "distal" was avoided in the complete manuscript.

4. How do the area fractions of porphyroclasts and matrix vary as a function of distance along the transect? This information might allow to better evaluate the continuity (or not) of the evolution of the deformation conditions along the transect.

> Descriptions and discussions of the variations for the abundances of porphyroclasts and the proportion of recrystallized matrix respectively are added in lines 392/ 440, 435/ 485 and 653/ 709.

5. How are the limits between porphyroclasts tails and the matrix defined? Is it really important to discriminate between these two microstructural domains?

> Neoblast tails of pyroxene porphyroclasts are characterized by a phase (pyroxene dominated, amphibole-bearing), grain shape (equaxial), grain size (coarse) composition and CPOs (AG- and B- type). All these microstructural characteristics differ distinctly from the surrounding mylonitic matrix with ol-dominated, mostly amph-absent composition, elongated grain shape, smaller grain size and strong A-type CPOs. Therefore, their limits are defined by all these microstructural parameters which enable an easy distinguishment between matrix and tails.

6. To discuss the variations in olivine CPO patterns along the transect as it is done lines 331-335 and 550-555, the full dataset needs to be presented in the article. It is stated in the text that A-type patterns dominate. Yet most figures presented in the main text show AG-type patterns.

> The complete CPO data was added as figure 4.

7. Deformation mechanisms in the matrix: please complete this point by showing and discussing the internal deformation of the neoblasts... If they deformed by dislocation creep, as stated in the ms., they should display, to some extent, a substructure (GNDs) consistent with this deformation.

> A GND reconstruction map for the mylonitic mixed matrix was added in figure 7 corroborating dislocation creep as dominant deformation mechanism

8. CPOs in the recrystallization tails: do the observations hint for inheritance of orientations from the porphyroclasts?

> Yes, there is a strong inheritance for opx neoblasts and a slightly weaker one for cpx. Description and discussion were added in lines 418/ 468, 464/ 515 and 677/ 734.

9. Line 558: What are the observations that indicate that deformation was enhanced by the presence of melts in the early stages of shearing? Why early stages?

For the discussion of the timing of the metasomatism please see above in the discussion of the main points of criticism. The potential effects of melt-presence in the studied rocks on the deformation and its microstructural implications are discussed in reworked section 5.2. The authors therein agree, that the initiation of strain localization cannot be conclusively attributed to the investigated metasomatic event. However, the comparison to other upper mantle shear zone shows a strong association of reactions, phase mixing and shear zones.

10. Line 559: Piezometric data cannot document the activation of a grain size sensitive mechanism. At best, given all the uncertainty and hypotheses inherent to this method, it allows an estimate of the active stresses. And this estimate is only valid if grain size reduction is controlled by dislocation creep, since these are the conditions prevailing in the experiments used for the calibration.

Correct, the paragraph was changed accordingly (ll. 722/ 779) .

11. Line 560: Where does the evidence for GBS is shown?

Evidence for GBS is tricky and in most cases no distinctive feature for GBS. Therefore, we refer to the research of Précigout et al. (2007) who argued for DisGBS as dominant deformation mechanism (ll. 724/ 782).

12. Not all pyroxenites in Ronda were interpreted as resulting from replacement of previous gt-pyroxenites by melt-rick reaction. An important volume of pyroxenites was interpreted as formed by partial (reactive) crystallization of percolating melts. Moreover, in the mylonites, gt-pyroxenites predominate (Garrido and Bodinier 1999).

Very correct. Thanks for the annotation, the text was changed accordingly (ll. 619/ 674).

13. Referencing is imprecise in some places. For instance, Passchier and Trow (1996) is not the best citation for viscoplastic anisotropy due to crystal orientation.

Citations have been checked and updated (e.g., l. 44/ 52). The authors would like to thank both reviewers for their paper suggestions.

14. Lack of cross-cutting relations between gt-mylonites and sp-tectonites was also reported by Soustelle et al (2009).

Thanks for the hint, the citation for Soustelle et al. (2009) was added (l. 128/ 145).

Minor comments / questions:

- In all figures presenting CPO the color bars indicating the intensities of the contours are missing.

Numbers of grains and color bars for ODFs were added to all orientation figures.

- In fig. 2 it is impossible to see the elongated opx porphyroclasts. Yet they are clearly visible in the field. Moreover, due to serpentinization and fine-grained nature of these peridotites,

to define in the field variations in mineralogical composition is very difficult. Is this figure really useful?

Figure 2 was dismissed.

- In figure 3, the microstructure of the tectonite is not visible.

Contrast of tectonite overview has been increased in figure 2.

- Line 150: the peridotite solidus is not a temperature, it depends on temperature and pressure, and composition, volatiles...

Of course. In this regard it is just a citation of the estimated T conditions. P conditions were added (l 152/ 176).

- What are the arguments (=observations) used to define an intergrowth between olivine and pyroxenes (line 313)?

Highly lobate boundaries to bordering olivines and weird shaped protrusions (see figure 7; l 333/ 379).

- What do the arrows in Fig. 12 mean? In Soustelle et al. (2009), only in two samples the analyzed pyroxenes were clearly identified as secondary, that is, resulting from partial crystallization from melts. In addition, the area concerned by this study is not indicated in Fig.1 as stated in the figure caption.

Arrow description was added in figure 13. In figure 1 area of Soustelle et al. (2006) was added.

---

## Author Response (AR2)

Dear Florian,

Thank you for your work on the manuscript and your fast processing. We fundamentally agree with all the points you have raised in your letter:

As the first review was very constructive a second opinion of Andrea would have been optimal for us, but even so she has her improving share in the manuscript.

In this second revision we focused on a clear separation of facts and hypotheses. On a structural level we divided more speculative parts of the discussion into separate sections (timing of the metasomatism (5.2.2) and the comparison to other upper mantle shear zones (5.5)). Additionally, we clearly indicated the limitations of this research in terms of geochemical data base and sampling density around the tectonite-mylonite transition whenever necessary. Finally, the complete discussion including the headings was rewritten to emphasize the main points and strengthen the argumentation. Furthermore, three research questions were formulated in the introduction which are answered by the trimmed bullet point conclusion.

Additional geochemical analysis would be desirable. The focus of this additional analysis, however, should lie on the above-mentioned tectonite-mylonite transition which would require another sampling campaign. Even though we fully agree with J. Précigout, this is unfortunately no longer feasible within the scope and possibilities of this project. However, in the more speculative part of the discussion, we want to point out the potential implications of the geochemical trends and overprinting relationships that may inspire future work exactly on this transition.

Regarding J. Précigout's second review, we had of course hoped that the corrective work done on the basis of Andreas and his review would also be constructive in his opinion. As one of his main points of criticism is the potential impact of melt on deformation, we propose a change in title to

"On the interaction of melt, phase mixing and deformation in a large-scale mantle shear zone (Ronda peridotite, Spain)"

which leaves more room for discussion. If you think that title is more convenient for the manuscript, the authors would agree on a change. The controversial parts J. Précigout addresses were moved to the more hypothetical sections of the discussion. The comparison of different upper mantle shear zones across different P-T conditions, Ronda included, indicates a significant role of reactions for the localization of strain in the upper mantle. This section does not aim on a proof for melt-enhanced deformation in Ronda. All his minor changes were edited (see below).

On behalf of the authors,
Sören

Minor comments/concerns by Jacques Précigout

Line 34 : delete « the » shear zones

   *Done.*

Line 49: Deformation-induced

   *Done.*

Line 53: Reaction-induced

   *Done.*

Line 69-70: Sorry but this sentence is wrong: we never proposed that stress increased where strain is localized. On the contrary, our model involves that strain localizes because of a stress drop due to grain size reduction. The difference of stress you see is related to the lithosphere strength profile.

   *Thanks for the remark. We changed the sentence accordingly (Line 65-66).*

Line 70: What does « mixed peridotites » means ? I think you are mentioning « mixed phases », but please, rephrase.

   *Paragraph was rephrased (Line 67-71).*

Line 202: ODF means Orientation Distribution function, not O. Density F.

   *Corrected.*

Line 200-215: By the way, there is something wrong there, because no one of your figure shows an ODF; there are only pole figures with or without texture, depending on the amount of data point. In your case, the ODF is only used to calculate the Jindex. Please, correct this paragraph accordingly.

   *Sentence was rephrased.*

Line 203: The Mindex is not based on the ODF, but on the relative distribution of uncorrelated misorientation angles between the measured and theoretical ones (Skemer et al., 2005).

   *Sentence was changed accordingly.*

Line 208: The color scale of pore figures refer to « multiple of UNIFORM distribution », not « random distribution ».

   *The color scale used in MTEX is based on the multiple of random distribution. At least as "mrd" function in MTEX itself and in the associated work flows (e.g. https://mtex-toolbox.github.io/ODFTutorial.html).*

Line 210: Please, expand on how you resolved the GND. Maybe you did it this way, but I don't think you have implemented the GND in MTEX directly from the paper of Pantleon (2008). I suppose instead you have used the functions « fitDislocationSystems », which has been implemented by the MTEX team. If I am correct, please modify your text accordingly.

   *Sentence was changed.*

Line 253: the same here: it seems that you confound ODF with pole figure. Please, clarify.

*Corrected. ODF was changed to pole figures.*

Figure 3: If you excludes the olivine-rich matrix from grain size calculations, which represents a significant part of the peridotites, how do you know that grain size does not reduce with increasing strain ? And what about the pinning effect on grain size ?

*To clarify, the dominant microstructure of all samples is the mixed matrix (Fig.2). In the mylonitic part, ol-rich matrix areas are only forming lenses in the finer-grained mixed matrix. It must be pointed out again, that in previous studies, ol-rich and mixed matrix have not been differentiated. As described in the text, ol-rich matrix domains are in fact rather limited in size and distribution and also "affected" by interstitial pyroxenes but just in a smaller amount than the mixed matrix. Phase mixing, smaller grain sizes, pinning by secondary phases and microstructures implying strong deformation are dominant in the mixed matrix. That is why we focus on the mixed matrix rather than on the olivine grain size of potentially annealed ol-rich matrix lenses. If the ol-rich matrix were of major importance for deformation one would not expect it to form competent lenses and rather interconnected layers like the mixed matrix (Fig. 2). The analysis of the ol-rich lenses could of course give grain size data (the data obtained indicates similar or bigger grain sizes than the mixed matrix), perhaps even smaller grain sizes towards the NW but as it is isolated in the mixed matrix grain size reduction or pinning in the ol-rich lenses does not seem decisive for the overall deformation. We did not exclude Ol-rich matrix domains from the start of this study, in fact it is the small abundance and it's strong serpentinization which hinders the analysis. Fortunately, this domain seems not to be decisive for the phase mixing and its influence on deformation. As the focus of this research lies on these aspects it is, from the point of the authors, legitimate and scientifically justifiable to exclude this domain.*

Figure 3: Furthermore, plotting the grain size data for all phases in one graph is difficult to read. Please, separate them, at least for grain size. To me, it seems that there is a slight reduction of the mean size in any case .

*Grain size data for all major phases was plotted separately.*

Line 311-312. Only giving one axis with respect to the foliation plane is not enough. You also have to say that [010] is normal to the foliation.

*[001] was given as it is indicative for deformation induced CPO for opx. [100] was added for clarification.*

Line 325: What is your percentage threshold in terms of secondary phases to say that you are dealing with a olivine-dominated matrix or a mixed matrix? Even for mixed matrix samples, most of the ones you described have olivine as the far dominant phase (up to 80%). Please, clarify.

*Ol-rich matrix is characterized by >90% olivine. The mixed matrix is defined by less than 90 % olivine and a secondary phase content dominated by interstitial secondary grains (pyroxenes). The latter characteristic is the more important one, as it points to the formation by melt infiltration. Thanks for the remark, this definition was added to the matrix section.*

Line 337: The phase abundance you describe does change with the strain gradient, so how melt can enhance strain localization?

> *Based on the microstructural results, the melt was apparently present in the entire shear zone. It rheologically affects therefore the entire melt affected area = all analyzed samples. The word "enhance" is probably wrong. The fact, that melt presence facilitates deformation does not have to be discussed here. Our point is that melt is present in the entire shear zone together with deformation indicative microstructures. Based on all obtained data and compared to Beni Boussera, it is harder to think these two points apart from each other, melt and deformation, than together. Renaming of the manuscript is considered together with the editor.*

Line 338: Coarse-grained olivines

> *Changed.*

Line 359: (mrd, M) ??? Please, give the values.

> *The sentence was rephrased. M and mrd values are mostly similar for both pyroxenes for a given microstructure. Values are given above and in the appendix.*

Line 407-410: in line 407, you say that grain size is constant through the whole transect, but in line 410, you mention that « in contrast to grain size, AR remain constant… ». Please, clarify.

> *Grain sizes do show an excursion towards bigger sizes around 250 m distance to the NW-B. This excursion is not present for AR. Sentence was rephrased.*

Figure 11: In terms of chemistry, no data has been added to confirm the chemical gradient. This is yet a major point of this paper.

> *We totally agree that this is a very interesting point. However, for a solid database more EPMA measurements would be necessary, as the reviewer pointed out in his first review. A solid evaluation of the trend in Mg# is not possible with the samples collected for this study. In this manuscript, the focus lies on the mylonites, their microstructures and genesis. Fortunately, the research has led us to these geochemical findings which, however, can only be investigated sufficiently by collecting additional samples from the tectonite-mylonite transition. This and the resulting geochemical focus are clearly beyond the scope of this manuscript.*

Line 684: You claim that grain size is different between tectonite and mylonite, but you say the opposite in the abstract. Please, clarify.

> *"The constant grain sizes with local variations independent on the distance the deformational centre indicate a broad scale deformation with nearly constant stresses in the entire mylonitic area." (now:"* Instead, extensive phase mixing under near steady-state conditions is documented by the constant grain size and by phase boundary percentages > 60% for the entire mylonitic unit and all the microstructural domains.") *The abstract is talking about the mylonitic part, not about a contrast between tectonites and mylonites. In the abstract, only the change in grain shape between tectonites and mylonites is adressed.*

Line 717: You say that there is a strong CPO along the transect, but in Précigout and Hirth (2014), we documented a decreasing CPO with increasing strain in the top mylonite. And this CPO get close to complete randomization nearby the shear zone boundary.

*A discussion on the "Strain localization in the northwestern mylonites" was added in section 5.2.3.*

Line 720: A-type is not dominant in the top mylonite, where AG-type and B-type mostly occur (Précigout and Hirth, 2014).

*A differentiation between top mylonites and mylonites is added. However, in the matrix we only found one B-type CPO. For the B-type genesis please see comment above, section on neoblast tails and CPO data.*

Line 734: Grain size reduction is a matter of neoblast amount, OK, but still, there is a grain size reduction with a possible rheological effect.

*Yes. A sentence was added in line 774-779 and the effect of smaller grain sizes by increased neoblast abundance is discussed in section 5.2.3.*

Line 745: De Ronde and Stünitz is not relevant here, because there is no plagioclase, nor other phase transition across the ronda shear zone (Spl stability field).

*We are referring to the mechanism of grains being transported from the reaction interface during deformation. The type of reaction should not be important for this process.*

Line 768: Yes, Tommasi is able to localize where melts are localized, which is apparently not the case here, based on what you show.

*The discussion was rearranged to get the main points clearer. The shear zone is where melt was present. This melt crystallized apparently under deformation as visible by the grain shape. Therefore, its presence apparently shaped the shear zone. We do not claim that it was there during the final stages of deformation, but it definitely formed the peridotites in regard to its mixed and reacted microstructure we examined in this manuscript.*

Line 793: What do you mean by « shaped the shear zone »?

*Please see the comment above.*